# Role of discrete water recharge from supraglacial drainage systems in modeling patterns of subglacial conduits in Svalbard glaciers

Léo Decaux[1], Mariusz Grabiec[1], Dariusz Ignatiuk[1], and Jacek Jania[1]

[1]Department of Geomorphology, Faculty of Earth Sciences, University of Silesia, 60 Bedzinska Street, 41-200 Sosnowiec

**Correspondence:** Léo Decaux (leodecaux@gmail.com)

**Abstract.**

As the behavior of subglacial water plays a determining role in glacier dynamics, it requires particular attention, especially in the context of climate warming that is increasing ablation and generating greater amounts of meltwater. On many glaciers, water flowing from the glacier's surface is the main source of supply to the subglacial drainage system. This system is largely influenced by the supraglacial drainage system that collects meltwater and precipitation and rapidly delivers it to discrete points in the glacier bed via moulins and crevassed areas, called water input areas (WIA). Models of patterns of subglacial conduits mainly based on the hydrological potential gradient without taking into account the supraglacial drainage system are still regularly performed. We modeled the pattern of subglacial channels in two glaciers located in Svalbard, the land-terminating Werenskioldbreen, and the tidewater Hansbreen during the 2015 melt season. We modeled a spatial and a discrete water recharge in order to compare them. First, supraglacial catchments were determined for each WIA on a high resolution digital elevation model using the standard *watershed* modeling tool in ArcGIS. Then, interpolated water runoff was calculated for all the main WIAs. Our model also accounts for several water pressure conditions. For our two studied glaciers, during the ablation season 2015, 72.5% of total runoff was provided by meltwater and 27.5% by precipitation. Changes in supraglacial drainage on a decadal timescale are observed in contrast to its nearly stable state on an annual timescale. Nevertheless, due to the specific nature of those changes, it seems to have a low impact on the subglacial system. Therefore, our models of subglacial channel are assumed to be valid for a minimum period of two decades and depend on changes in the supraglacial drainage system. Results showed that for Svalbard tidewater glaciers with large crevassed areas, models of subglacial channels that assume spatial water recharge may be somewhat imprecise but are far from being completely incorrect, especially for the ablation zone. On the other hand, it is important to take discrete water recharge into account in the case of land-terminating Svalbard glaciers with limited crevassed areas. In all cases, considering a discrete water recharge when modeling patterns of theoretical subglacial channels seems to produce more realistic results according to current knowledge.

# 1 Introduction

In the context of global climate change and in particular, the rapid melting of glaciers around the world, it is essential to understand changes in their meltwater drainage system and its consequences for glacier behavior. Today it is even more important to focus on the hydrological system of Arctic glaciers given that the Intergovernmental Panel on Climate Change (IPCC) predict

longer summer seasons (Pachauri et al., 2014) and also knowing that Svalbard glaciers have been shrinking for several decades already (Błaszczyk et al., 2013; Hagen et al., 2003b). All predictions assume an increase in runoff (meltwater and precipitation) from Arctic glaciers, ice caps and ice sheets, suggesting intensification of their whole drainage system and consequently of their dynamics and their impact on sea level rise (Hagen et al., 2003a; Hanna et al., 2008; Mair et al., 2002; Nuth et al., 2010; Sundal et al., 2011). Meltwater and precipitation water are directly linked to glacier dynamics by supplying the subglacial

drainage system which lubricates the ice-bed interface thereby reducing the basal shear stress resisting ice flow (Bartholomew et al., 2012; Hoffman et al., 2011; Shepherd et al., 2009). In the case of tidewater glaciers, an increase in velocity means an increase in calving rate and hence a higher loss of mass thereby contributing even more to sea level rise. While Greenland and Antarctica are currently considered to be the main future players controlling sea level rise (DeConto and Pollard, 2016; Price et al., 2011; Rignot et al., 2011), it is crucial to understand how supraglacial, englacial and subglacial drainage systems

influence each other in a glacier system, as this knowledge will make it possible to improve ice sheet models.

Nowadays, several models of subglacial conduits do not take the supraglacial drainage system into account (Fischer et al., 2005; Grabiec et al., 2017; Hagen et al., 2000; How et al., 2017; Pälli et al., 2003; Sharp et al., 1993; Shreve, 1972; Willis et al., 2009). As a result, they consider a spatial recharge (the term recharge is used here to refer to meltwater and precipitation water entering in the subglacial drainage system from the surface of the glacier) meaning that the water recharge is homogeneous

or with some local water values over the entire surface of the glacier. This is one of the biggest assumptions that leads to inaccurate modeling of the locations of subglacial channels (Gulley et al., 2012).

However, due to their direct impact on the englacial and subglacial drainage system, studying supraglacial drainage systems is vital. Knowledge of these systems makes it possible to locate where the supraglacial system switches into an englacial and then subglacial system via moulins, shear fractures or crevasses, called water input areas (WIAs), and hence to better estimate their

water recharge (Bartholomew et al., 2011; Benn et al., 2017). Indeed, concentrated surface water streams are necessary for the formation of a channelized internal drainage system (Mavlyudov, 2006). Likewise, the supraglacial drainage system largely influences the subglacial system by collecting meltwater and precipitation water and rapidly delivering it to discrete points in the glacier bed via WIAs (Catania and Neumann, 2010; Gulley, 2009; Poinar et al., 2015; Smith et al., 2015). In addition to being an important source of water for the internal drainage system of glaciers, part of the englacial conduits are formed by

direct incision of supraglacial channels followed by creep closure (Gulley et al., 2009; Irvine-Fynn et al., 2011). Nevertheless, the supraglacial drainage system remains one of the least studied hydrologic processes on Earth (Smith et al., 2015).

Spatial recharge is theoretical and is not actually observed on any glacier. Consequently, some models have used discrete recharge but only refer to limited areas on a glacier or with randomly placed moulins or even as a model validation method (Hewitt, 2013; Werder et al., 2013). In fact, no precise representation of the entire supraglacial drainage system (including

inputs from both crevasses and moulins) has been used to constrain a subglacial channels model. Finally, no comparison study on the consequences of assuming spatial recharge has been performed.

Our study focused on the land-terminating glacier Werenskioldbreen and the tidewater glacier Hansbreen, both located in the southern part of the Svalbard archipelago and representative of many Svalbard glaciers (Fig.1).

The first step consisted in identifying and mapping the supraglacial drainage features of the two glaciers using very high resolution remote sensing images for the years 1990, 2010, 2011 and 2015. Combination with field observations allowed us to locate water flows inside the glaciers via moulins and crevasses (Benn et al., 2017, 2009; Holmlund, 1988; Nienow and Hubbard, 2006; Van der Veen, 2007). In the second step of this study, we calculated the catchment area of each main WIA on both glaciers. Subsequently we estimated the total amount of surface water (precipitation and meltwater) on the whole surface

area of the two glaciers together with their surrounding slopes for the entire 2015 ablation season. This allowed us to map WIAs location along with their absolute water recharge volumes. The final step consisted in modeling the pattern of subglacial conduits in the two glaciers using spatial recharge and discrete recharge for comparison.

Our attempt should improve current modeling of the theoretical pattern of subglacial channels. More specifically, we discuss our results with the subglacial conduit models proposed by Pälli et al. (2003) and Grabiec et al. (2017) for our two study

glaciers, as these are only based on the hydrological potential gradient (Fischer et al., 2005; Flowers and Clarke, 2002; Sharp et al., 1993; Shreve, 1972). Moreover, contrasting model results with spatial and discrete water recharge enables a better understanding of the influence of this parameter for all glaciers.

## 2 Study sites and datasets

### 2.1 Study sites

Two different types of polythermal Svalbard glaciers were chosen because their morphology, surface and subglacial topography, dynamics, thermal state and hydrology are representative of important types of Svalbard glaciers (Grabiec et al., 2012b; Hagen et al., 1993, 2003a; Ignatiuk et al., 2014). Werenskioldbreen is a land-terminating glacier and Hansbreen a tidewater one. They are characterized by two different dynamics that have a direct impact on changes in both the surface topography and the drainage system. Werenskioldbreen is located in south-west Spitsbergen (77°05' N, 15°15' E) (Fig.1), flows from east to west

with an average speed of less than 10 m yr$^{-1}$ in two parallel flows divided by a central moraine (Baranowski, 1970; Grabiec et al., 2012a). It is a quite small Svalbard glacier, 6.5 km long, 2.2 km wide, with a surface area of 27.1 km$^2$ between 40 and 650 m a.s.l (Ignatiuk et al., 2014; Majchrowska et al., 2015). It has a maximum thickness of about 275 m and a cold ice snout less than 50 m thick frozen to the bedrock up to 700 m upstream from the front line (Navarro et al., 2014; Pälli et al., 2003). Its entire surface is composed of cold ice (below the pressure melting point) overlying a temperate ice layer (at the pressure

melting point) (Grabiec et al., 2017), thereby enabling the presence of a well-developed supraglacial drainage system (Ryser et al., 2013). Hansbreen, located at the entrance of Hornsund (77°04' N, 15°38' E) (Fig.1), flows north to south with a velocity

of c. 150 m yr$^{-1}$ at the front and of 55-70 m yr$^{-1}$ 3.7 km upstream (Błaszczyk et al., 2009). Situated between 0-664 m a.s.l, it is a medium size Svalbard glacier, 15.6 km long, c. 2.5 km wide on average with a surface area of 53 km$^2$. Its terminus forms a c. 30 m high cliff 1.9 km in width ending directly in the sea (Błaszczyk et al., 2009). Its mean ice thickness is 171 m and its maximum ice thickness is 386 m (Grabiec et al., 2012b). Temperate ice, firn and snow are present in the large accumulation

5    area, allowing water to percolate down to the surface of the temperate ice and preventing the formation of supraglacial channels. The structure of the ablation area is similar to that of Werenskioldbreen with a cold ice layer overlying temperate ice (Grabiec et al., 2017; Navarro et al., 2014), preventing dispersed infiltration of the water below the Equilibrium Line Altitude (ELA). However, most of the surface is crevassed, thereby limiting the development of a supraglacial drainage system. Crystal Cave and Bird Brain Cave located on Hansbreen (Fig.1) are the only two well-studied and well mapped intra-glacial drainage systems

10   in our study area (Gulley et al., 2012; Murray et al., 2007; Schroeder, 1998; Turu, 2012).

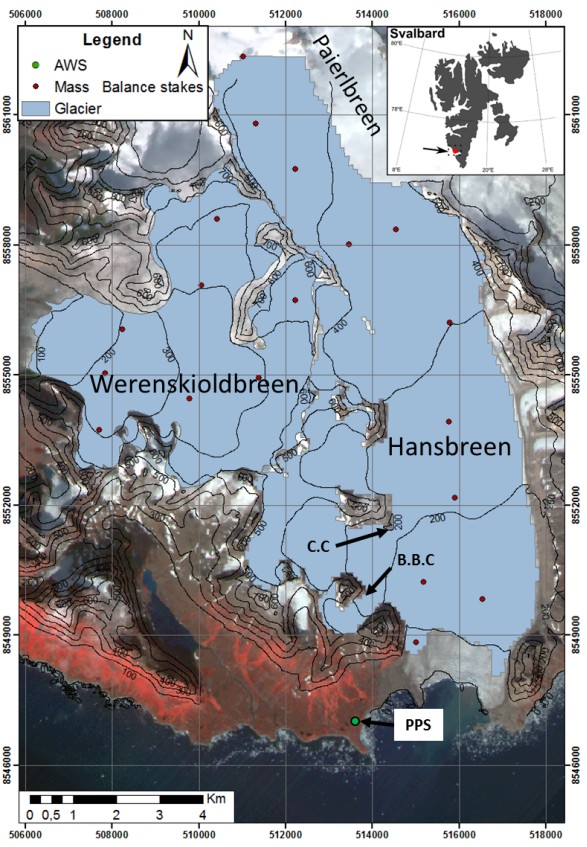

**Figure 1.** Map showing the location of Hansbreen and Werenskioldbreen, Svalbard, the automatic weather station (AWS), Polish Polar Station (PPS), Mass balance stakes network, Crystal Cave (C.C) and Bird Brain Cave (B.B.C). The background map is a SPOT satellite image acquired on 16/08/1988 and the coordinate system used is WGS 1984 UTM zone 33N.

## 2.2 Datasets

We used very high resolution (VHRS) WorldView-2 satellite images, with 0.5 m resolution, acquired on 21/08/2015 and Geo-eye satellite images acquired on 10/08/2010. We also worked with sets of Norwegian Polar Institute aerial photos from 1990 and 2011. In addition to remote sensing data, field observations and a Global Positioning System (GPS) survey, were necessary
to identify control points to calibrate mapping. Maps of the supraglacial drainage system already exist for Werenskioldbreen in 1990 and 2010 (Ignatiuk, 2012; Pulina et al., 1999). Bedrock digital elevation models (DEM) of Hansbreen and Werenskioldbreen at a spatial resolution of 100 m and a vertical resolution estimated at $+/-5$ m, were obtained during a survey conducted in April 2008 by the University of Silesia and Institute of Geophysics Polish Academy of Sciences teams, combining GPS/GPR (ground penetrating radar) measurements (Grabiec et al., 2012b). We also created a high resolution surface
DEM of both glaciers, using WorldView-2 VHRS images for the year 2015. Finally we used a meteorological dataset from the automatic weather station (AWS) located at the Polish Polar Station (PPS) (about 1.5 km from Hansbreen front) to calculate the volume of water produced by melt and precipitation during the 2015 melt season (Fig.1).

## 3 Methods

### 3.1 Mapping supraglacial drainage

Based on high resolution images, we generated several maps of the supraglacial drainage system of the two glaciers for different years using ArcMap software.

We mapped the surface streams manually, leading to personal choices and naturally to some subjective decisions. We decided to map only active streams with a minimum width of one meter, knowing from field observations that numerous smaller streams are present, along with the limitation due to the spatial resolution of the VHRS images. In addition, we manually
mapped crevassed areas and moulins with a diameter greater than one meter, enhanced by direct field observations and GPS measurements.

### 3.2 Calculation of WIA catchments

First, based on the supraglacial maps (section 3.1), we created 2015 WIA maps for both glaciers (Fig.2). WIAs were defined as substantial crevassed areas or crevassed areas intersecting active surface streams and groups of active moulins. In the large
accumulation area of Hansbreen, water percolates through the snowpack, then through the firn, to finally create a layer of saturated water at the temperate ice/firn interface (Fountain and Walder, 1998; Lliboutry, 1971). The water flows along this interface, comes to the surface at the ELA or reaches the englacial system of the ablation area thanks to crevasses in the accumulation zone. Because the area situated just below the ELA is considered as a WIA (large crevassed zone), we included the water from the accumulation area in it.

Next, the WorldView-2 stereo pair image from 2015 was processed with Geomatica software to create a surface DEM of our study area. The resulting DEM with a 4 m spatial resolution was delineated for both glaciers thanks to contour files obtained from orthorectified WorldView-2 images with a vertical accuracy of $+/-1.5$ m. We filled in the small sinks on the DEM caused by small imperfections that occurred while we were creating the DEM, giving the impression small lakes formed on the surface which has never been observed in the second part of the ablation season on either glacier. From the corrected DEM, we calculated the flow direction from each pixel to its steepest downslope neighbor.

Finally, using standard *watershed* modeling tool in ArcGIS, we determined the water catchment area of each WIA with 4 m spatial resolution (Fig.2). Some manual adjustments of the catchments delineation were made when necessary. The most common correction was to extend the catchment area where an active stream ending in a WIA was not included in it.

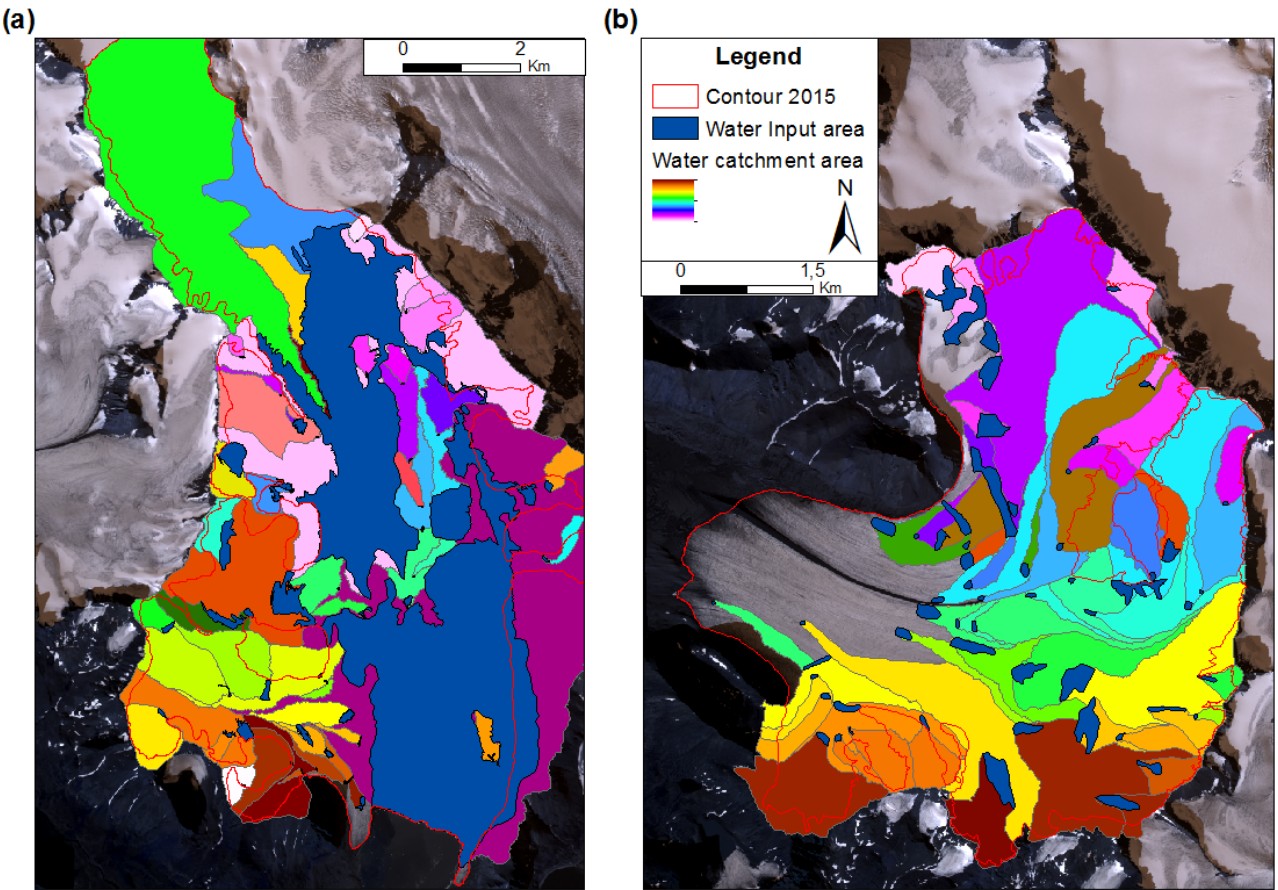

**Figure 2.** Hansbreen (a) and Werenskioldbreen (b) WIA catchments in 2015. The background of the map is a WorldView-2 VHRS image acquired on 21/08/2015.

### 3.3 Estimation of spatially distributed runoff

The main subglacial water source is known to be controlled by runoff water from the surface of the glacier, mainly surface meltwater and precipitation (Flowers and Clarke, 2002; Hodson et al., 1998; Irvine-Fynn et al., 2011). Therefore, to create a quantitative subglacial water flow model, we extrapolated the amount of meltwater generated by the entire surface of the glacier and the amount of the total precipitation (solid, mixed and liquid) falling on the entire surface of the glacier and on the surrounding slopes for the 2015 melt season (06/06/2015 to 10/10/2015). Considering all types of precipitation allows us to take into account the potential meltwater originating from summer accumulation (due to solid and mixed precipitation). In fact, we assumed that all snow deposited during the ablation period will melt due to positive average temperature over this season. Regarding the large accumulation area of Hansbreen, a large part of the water stored in the firn due to internal accumulation was included in our model. Thanks to a study by Grabiec et al. (2017), we disposed of an estimation of the refreezing of capillary and percolation meltwater (excluding capillary water that freezes in the fall) in the snow and firn. It was estimated that in season 2007/2008 38% of the total meltwater refroze in the snow and firn located above the ELA. Therefore, in our calculations, we considered that 38% of the total meltwater in this area was trapped within the firn corresponding to 5.5 $10^6$ m$^3$ for the year 2015.

Spatial distribution in the surface ablation model was generated based on the summer mass balance data provided by the World Glacier Monitoring Service (WGMS), relying on the mass balance stake network present on both glaciers (Fig 1) (Błaszczyk et al., in press). WGMS provides data about accumulation and ablation points on the glaciers (WGMS, 2017). Surface ablation data combine meltwater produced by the winter snow cover at the beginning of the melt season, and glacier surface melt during the rest of the melting period. The relationship identified between the summer mass balance and elevation ($R^2$= 0.83) allowed us to model meltwater production ($Q_M$) for the whole glacier. Surface ablation was approximated by interpolation of stake data over the area (grid 100x100m) at a range of elevations.

In the precipitation model, spatial distribution was calculated as follows. Applying a precipitation gradient ($\Delta_P$) of 19% per 100 m (including catching error) calculated by Nowak and Hodson (2013) to the precipitation measured at PPS ($Q_0$), we were able to calculate the amount of precipitation ($Q_P$) at each altitude ($h$), in any DEM cell using 1:

$$Q_P = Q_0 + (\Delta_P Q_0 h) \tag{1}$$

Therefore, combining the meltwater produced by the whole glacier with the amount of precipitation at each altitude, we were able to calculate the total glacier runoff ($Q_H$) in any DEM cells following equation 2:

$$Q_H = Q_M + Q_P \tag{2}$$

Surface ablation modeling errors ($\sigma_{Q_M}$) were calculated using the standard error of the regression. Errors in the spatial distribution of the precipitation model ($\sigma_{Q_P}$) were calculated using the method of total differential function according to equation 3:

$$\sigma_{Q_P} = (\Delta_P h + 1)\sigma_{Q_0} + \Delta_P \sigma_h Q_0 \tag{3}$$

where $\sigma_h$ is the DEM error and $\sigma_{Q_0}$ the precipitation measurement error at the PPS.

Therefore, we were able to calculate the total glacier runoff error $\sigma_{Q_H}$ according to equation 4:

$$\sigma_{Q_H} = \sqrt{\sigma_{Q_M}^2 + \sigma_{Q_P}^2} \tag{4}$$

We are aware that the error of precipitation spatial distribution is possibly larger due to expected substantial meteorological variations between rainfall events. However, calculated total glacier runoff error coincide with Nowak and Hodson (2013)
estimations. Moreover, it should be kept in mind that we might have underestimated the total amount of water runoff due to storage of liquid water in the snowpack and in the firn layer during the winter/spring period which is released during the melt season (Arnold et al., 1998).

### 3.4   Subglacial modeling

The theoretical pattern of subglacial channels was modeled for the year 2015. This required knowledge of the surface and
bedrock topography of the glacier (section 2.2), and of the spatial distribution of ice thickness resulting from them. The spatial resolution of our model was limited by the 100 m resolution of the bedrock DEMs. We consequently had to upscale surface DEMs (section 2.2), WIA catchment maps (section 3.2) and the spatially distributed water runoff model (section 3.3) in a 100 m grid. Surface and bedrock DEMs had a vertical accuracy of a few meters, which does not have much impact on our results with respect to the resolution of the spatial model (Fischer et al., 2005).

Water is known to circulate on, in and under glaciers in response to the hydraulic potential gradient (Shreve, 1972). Many models, and particularly the latest theoretical pattern of subglacial channels modeled by Grabiec et al. (2017); Pälli et al. (2003), are based on this gradient. We also based our model on this gradient except that we considered water circulation depends not only on the hydraulic potential gradient but also on some glaciological components. Subglacial drainage patterns can be modelled by assuming a spatially uniform flotation fraction $K$, which is the ratio between water pressure ($P_w$) and ice
overburden pressure ($P_i$) (Flowers and Clarke, 1999) according to equation 5:

$$K = \frac{P_w}{P_i} \tag{5}$$

Therefore, as gridded values of surface and bedrock elevation can be used to model the subglacial drainage pattern from calculated hydraulic potential field $\Phi$, we used equation 6:

$$\Phi = \rho_w g z_b + K[\rho_i g(z_s - z_b)] \tag{6}$$

where $\rho_w$ is water density (1000 kg.m$^{-3}$), $\rho_i$ is ice density (917 kg.m$^{-3}$), $g$ is the acceleration due to gravity (9.81 m.s$^{-2}$), $z_b$ and $z_s$ are respectively bed and surface elevation.

The direction of subglacial water flow was determined based on the hydraulic potential field ($\Phi$ calculated for each grid cell). Water flows perpendicularly to the equipotential lines of the hydraulic potential field. We calculated the flow direction in each cell by identifying the neighboring cell with the lowest hydraulic potential value. The next step in the simulation calculates accumulated flow into each grid cell according to our flow direction model. The grid cells with the highest accumulation shape the lines of preferred water flow. Several models stop at this stage, for which the cell value denotes a cumulative number of cells due to the water inflow to this specific cell. Such a model may be able to successfully predict the location of plumes in front of a tidewater glacier (How et al., 2017). In order to go one step further, knowing the size of the cell, the value can easily be transferred to a drained surface area. In order to quantify the water drained through the system, the total amount of meltwater and precipitation ($Q_H$) in the 2015 melt season was calculated and assigned to each grid cell, as described in section 3.3. The cells' values were then accumulated as described above, giving concentrated flow lines and water values through specific cells.

## 3.5 Model runs

We created two different simulation scenarios to take some glaciological and meteorological components into account:

1. **Hydraulic potential (ablation + precipitation input)**
   This model considers a spatial recharge of water, all the grid cells of the model are weighted as a function of the spatially distributed water runoff model values ($Q_H$) (section 3.3) (Fig.3). It corresponds to the last stage of the theoretical pattern of subglacial channels models achieved in our study area by Grabiec et al. (2017); Pälli et al. (2003) but updated for the year 2015 and with water volume values.

2. **WIA (ablation + precipitation input)**
   This model considers a discrete recharge of water, all the grid cells of the model corresponding to a WIA are weighted by the total amount of runoff ($Q_H$) (section 3.3) occurring on their particular catchment (section 3.2). All the other grid cells of the model are weighted with a value equal to 0 (Fig.4). (Furthermore, in order to study the proportion of melt and precipitation water, the model has been run considering either the amount of meltwater ($Q_M$) or the amount of precipitation ($Q_P$)).

Accordingly for scenario (2), the volume of water reaching each moulin or crevasse area was calculated. It depended not only on the surface topography but also on the surface conditions such as bare ice, firn and snow.

Water pressure in conduits depends directly on discharge (Röthlisberger, 1972) that in turn, relies on recharge. Therefore, water pressure in conduits is directly affected by the available amount of surface water (melt and precipitation) and Majchrowska et al. (2015) observed marked fluctuations in melt and precipitation rates during the ablation seasons from 2007-2012 on Werenskioldbreen. High pressure events (water pressure at ice overburden pressure) have been observed in the internal drainage system of Hansbreen at the beginning of intense melting periods (mainly in June and July) (Benn et al., 2009; Pälli et al., 2003; Schroeder, 1998; Vieli et al., 2004); and artesian features and over pressurized water outflows have been observed on Werenskioldbreen (Baranowski, 1977). Consequently, we modeled the subglacial channels for different $K$ values ($K = 1$; $K = 0.85$; $K = 0.75$; $K = 0.5$; $K = 0.25$; $K = 0$) for the two different scenarios for both glaciers, resulting in 24 simulations.

Therefore, our model considers:

(i) The surface properties of the glacier, and hence the location of runoff and water percolation areas;

(ii) The supraglacial drainage catchment structure of the glacier with respect to the WIAs, and hence the volumes of runoff along particular drainage pathways;

(iii) The water volume (meltwater plus precipitation) input to the system throughout the ablation season;

(iv) Several water pressure $K$ scenarios in the channels, mainly to illustrate distinct periods of the melt season.

In theory, a subglacial drainage system of glaciers involves a distributed and channelized system (Kessler and Anderson, 2004). Because most subglacial water transport occurs in conduits, sustained by the balance between the creep closure effect and melting due to heat released by the water flux (Hewitt, 2011; Nye, 1953; Röthlisberger, 1972), we did not include the distributed part of the subglacial drainage system but focused on the channelized component.

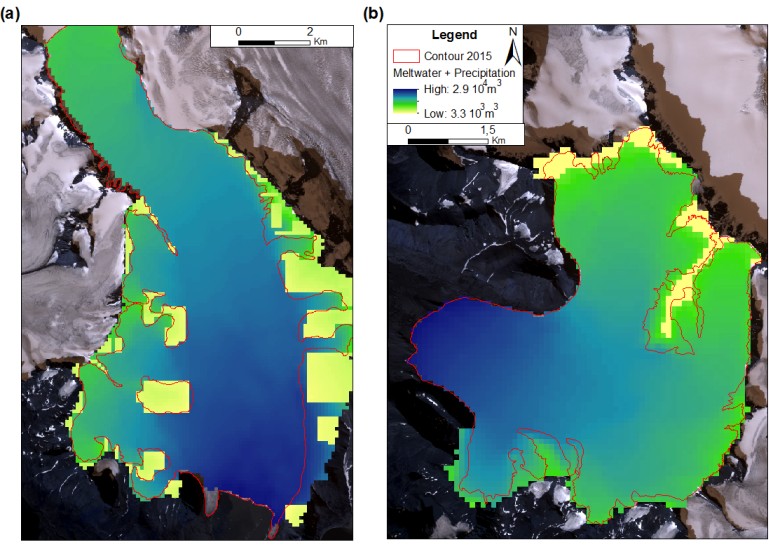

**Figure 3.** Spatial water recharge (meltwater + precipitation) for Hansbreen (a) and Werenskioldbreen (b) in 2015. The map background is a WorldView-2 VHRS image acquired on 21/08/2015.

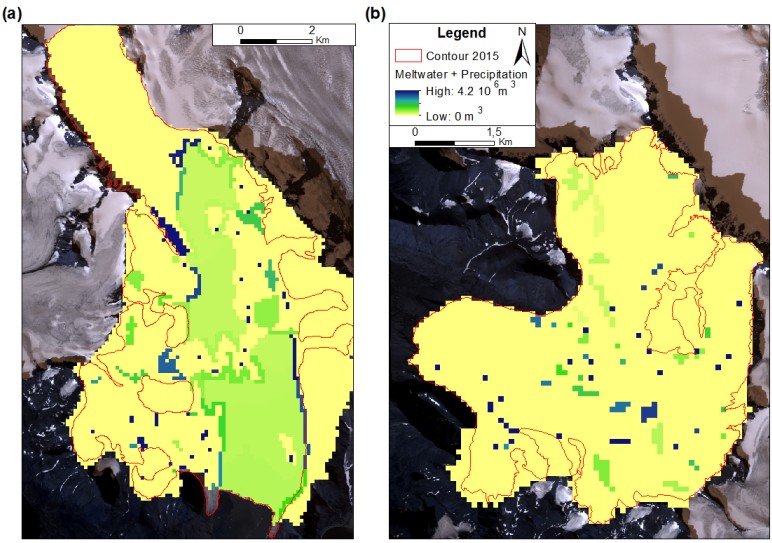

**Figure 4.** Discrete water recharge (meltwater + precipitation) for Hansbreen (a) and Werenskioldbreen (b) in 2015. The map background is a WorldView-2 VHRS image acquired on 21/08/2015.

## 4    Results

### 4.1    Temporal changes in the supraglacial drainage system

The mapping of the supraglacial drainage systems of the two glaciers in 2015 only included crevassed areas, moulins, superficial percolation zones and runoff places leading to the WIAs used in our model (Fig.2). However, maps of the whole supraglacial drainage system (including surface streams) for different years were only produced for Werenskioldbreen. In fact unlike Hansbreen, it has limited crevassed areas sustaining a well-developed supraglacial drainage system that allowed us to study the changes it has undergone.

We compared the supraglacial drainage system of Werenskioldbreen at two different timescales, annual (2010-2011) and decadal (1990-2010). Changes in the glacier's geometry were observed in this period (Gajek et al., 2009; Ignatiuk et al., 2014). The supraglacial hydrology of glaciers depends directly on the surface topography (Grabiec et al., 2012b; Nienow and Hubbard, 2006). A change in the supraglacial drainage system would therefore be expected at the decadal timescale, giving us the opportunity to better understand the physical mechanisms controlling it. Figure 5(a) shows the different supraglacial drainage features for the years 1990 (black) and 2010 (blue). First, the fact the surface streams are consistent in the two years is clearly visible, especially on the lower part of the glacier. Then several changes in the system can be observed, identified by numbers in Figure 5(a):

1. Creation of new moulins deactivating downstream surface streams.

2. Occurrence of new crevasses or shear fractures deactivating downstream surface streams.

3. Abandoned moulins due to their flowing out of a depression area, because of glacier's motion, or due to the deactivation of upstream surface streams.

4. It was impossible to map the surface drainage features due to a thick snow cover at the end of the 2010 ablation season.

Figure 5(b) shows the different supraglacial features in the years 2010 (blue) and 2011 (green). The two supraglacial drainage systems are more consistent, and some small differences can be distinguished due to the snow cover which made it impossible to map exactly the same areas.

### 4.2    Modeling the theoretical pattern of subglacial channels

The results of the simulations of scenarios (1), mentioned in section 3.4, represents the first improvement to the model of our study area mentioned in this article: we now have a quantitative model compared to the previous models of Grabiec et al. (2017) and Pälli et al. (2003) which are only qualitative.

Because all the simulation results for $K < 0.85$ display almost the same subglacial conduits pattern and clear differences are visible between results for $K = 0.85$ and $K = 1$, we only consider three water pressure states of simulations in this study,

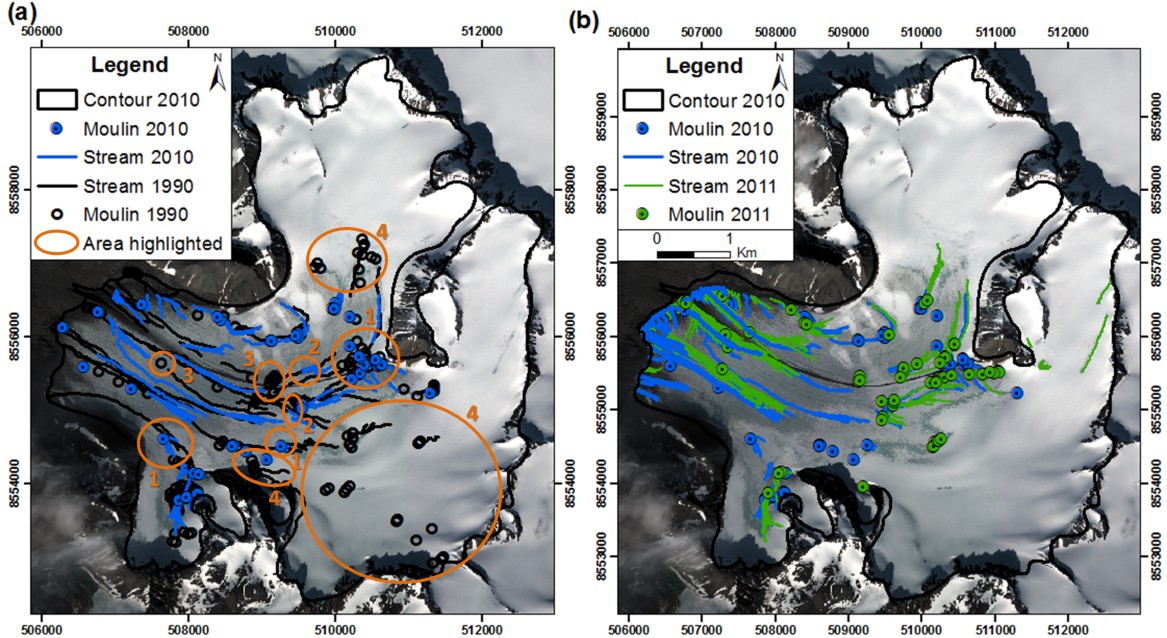

**Figure 5.** Comparison of the supraglacial drainage system for Werenskioldbreen on a decadal timescale (a) and on an annual timescale (b). Explanations for the numbers are given in in the text. The map background is a GeoEye-1 VHRS image acquired on 10/08/2010 and the coordinate system used is WGS 1984 UTM zone 33N.

$K = 1$, $K = 0.85$ and $K < 0.85$, represented here by $K = 0$ (Fig.6 and 7). The three other states $K = 0.25$, $K = 0.5$ and $K = 0.75$ are given in the appendix section (Fig.1 and Fig.2).

Spatial and discrete water recharge maps (Fig.6) showed very similar results for Hansbreen except that Crystal Cave and Bird Brain Cave conduits were better represented with a discrete water recharge. In the simulations, no subglacial channels started at these two locations, as is the case in reality when considering a spatial water recharge. In all cases, a general north-west to south-east subglacial water flow exists with one main channel in the eastern part of the glacier. Near the glacier front, the main flow direction changes from north-east to south-west. In the simulations in which $K = 0$ (Fig.6(c) and Fig.6(f)), all the channels are connected to the main channel and have the same outflow at the glacier front. Simulations in which $K = 0.85$ (Fig.6(b) and Fig.6(e)) and $K = 1$ (Fig.6(a) and Fig.6(d)) show three outflows, the two eastern ones are consistent in the simulations, but the western one is located further west when $K = 1$ than when $K = 0.85$ (Fig.6(a) and Fig.6(d)). In all the scenarios, except scenario (1) with $K = 0$, the main subglacial channel is generated just below the firn line (Fig.6).

Concerning Werenskioldbreen, compared to scenario (1) (Fig.7(a); 7(b); 7(c)), scenario (2) (Fig.7(d); 7(e); 7(f)) displays a more dendritic channel network in the central part of the glacier and the conduits start at lower elevations. All the simulations show a main channel flowing in the central part of the glacier, with the same outflows when $K = 1$ (Fig.7(a) and Fig.7(d)) and

$K = 0$ (Fig.7(c) and Fig.7(f)), and an outflow located further south when $K = 0.85$ (Fig.7(b) and Fig.7(e)). Scenario (1), in which $K = 1$, shows five outflows (Fig.7(a)) whereas in scenario (2) three outflows are modeled (Fig.7(d)). In the simulations in which $K = 0.85$, three outflows are visible in scenario (1) (Fig.7(b)) and two outflows in scenario (2) (Fig.7(e)). When $K = 0$, both scenarios exhibit only one outflow (Fig.7(c) and Fig.7(f)). Compared to the previous model proposed by Pälli et al. (2003), none of our model scenarios suggest a subglacial flow separated by the medial moraine present on Werenskioldbreen.

Overall, compared to spatial recharge results, discrete recharge maps exhibit conduits starting at lower elevations with additional subglacial branches, meaning they match the location of the moulins and small crevassed areas better.

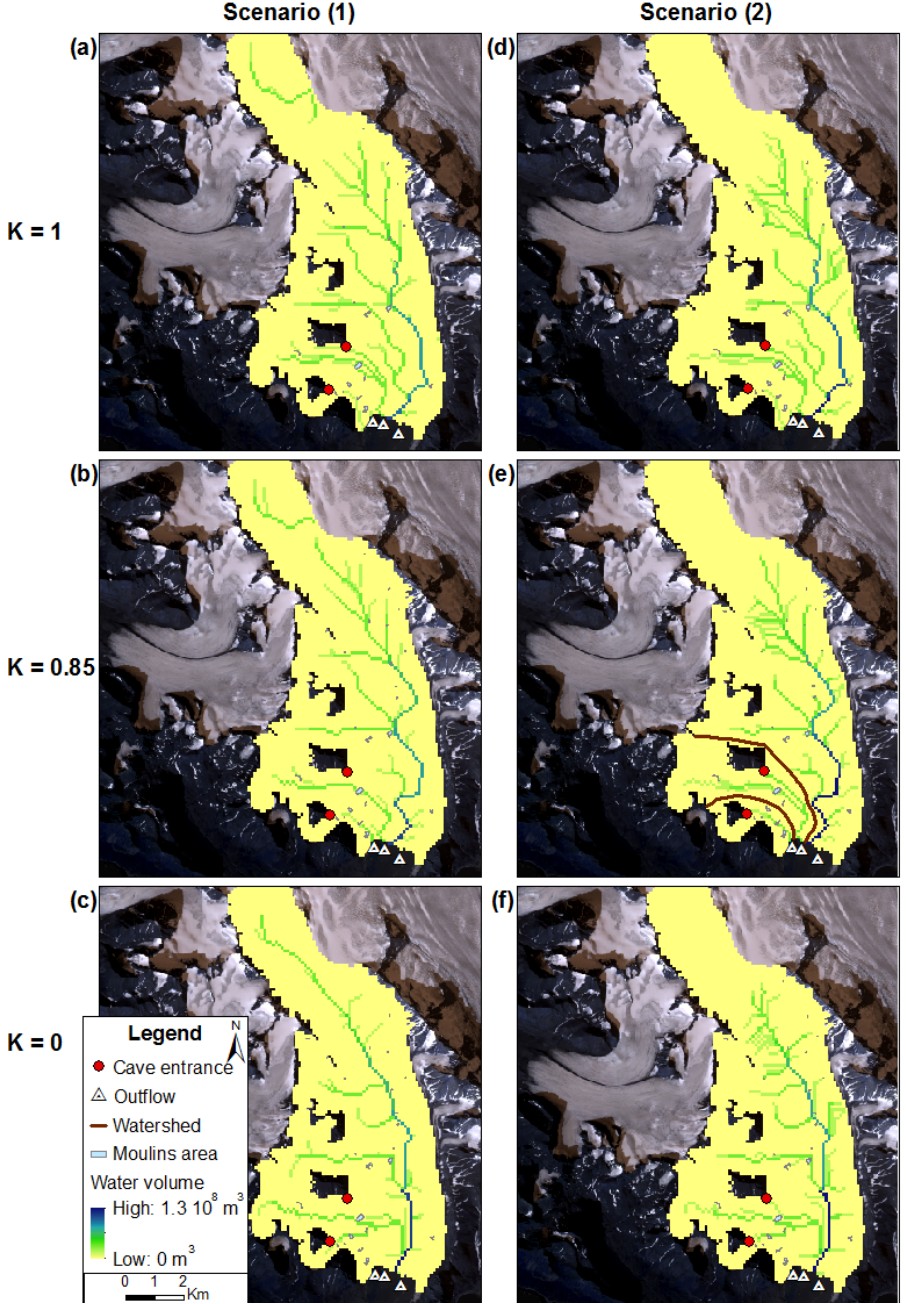

**Figure 6.** Map of the theoretical pattern of subglacial channels of Hansbreen modeled with scenario (1) ($K = 1$ (a); $K = 0.85$ (b); $K = 0$ (c)) and (2) ($K = 1$ (d); $K = 0.85$ (e); $K = 0$ (f)). The map background is a WorldView-2 VHRS image acquired on 21/08/2015.

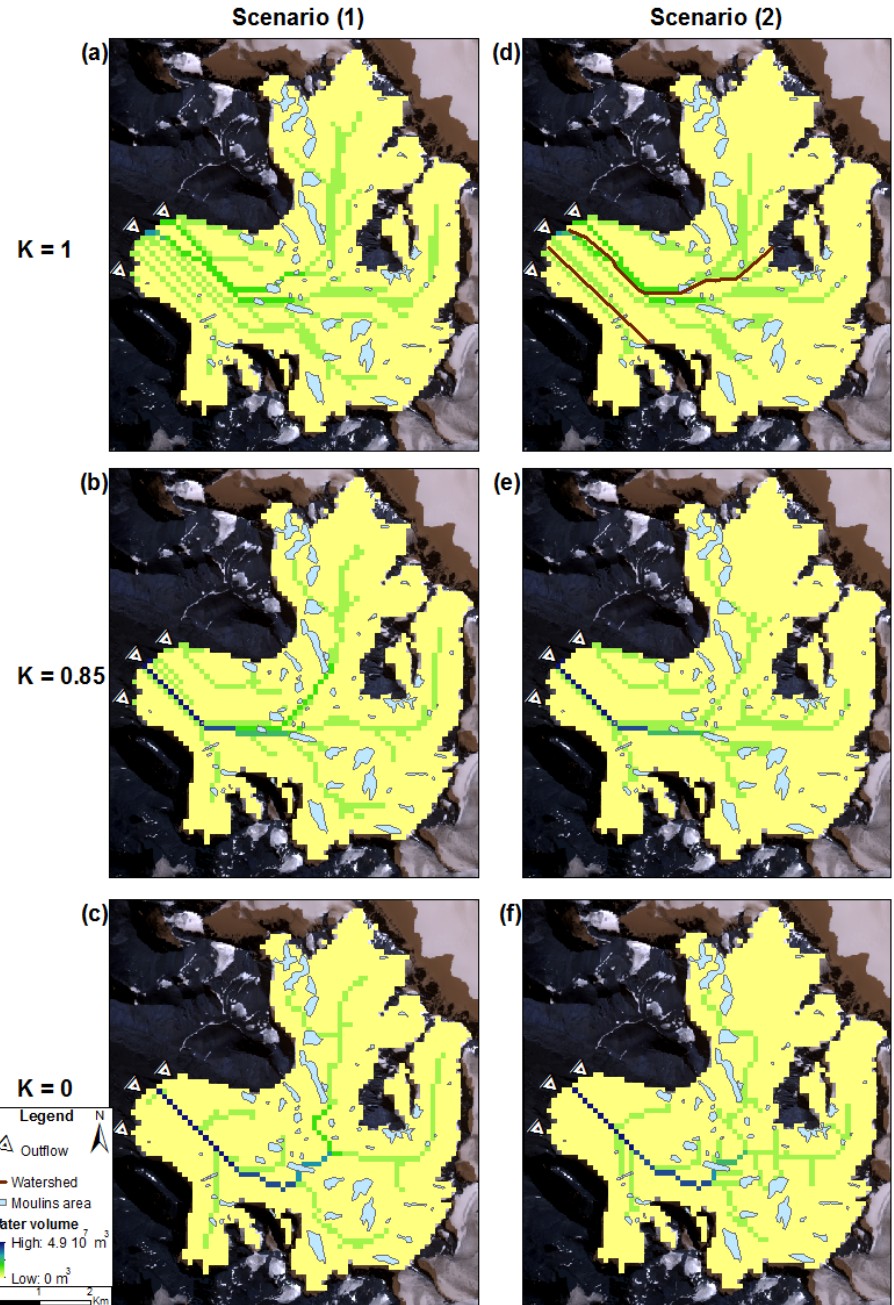

**Figure 7.** Map of the theoretical pattern of subglacial channels in Werenskioldbreen modeled with scenario (1) ($K = 1$ (a); $K = 0.85$ (b); $K = 0$ (c)) and (2) ($K = 1$ (d); $K = 0.85$ (e); $K = 0$ (f)). The map background is a WorldView-2 VHRS image acquired on 21/08/2015.

# 5 Discussion

We attempted to automate surface stream mapping, as already achieved for Greenland by Yang and Smith (2013), using a specific normalized difference water index for the ice surface ($NDWI_{ice}$). $NDWI_{ice}$ uses a normalized ratio of blue and red bands that allows each glacier pixel to be classified as either "water" or "non-water". The fact that it was not successful has several explanations. First, supraglacial streams in Svalbard glaciers are much smaller than those in Greenland. Second, Svalbard glaciers are surrounded by mountains that influence the surface conditions of the glaciers, which is not the case for the Greenland ice sheet. The surfaces of Svalbard glaciers are "dirty", i.e. they contain many small rocks or dust that blows down the mountain slopes and lands on the surface of the glaciers, changing the reflectance properties of the surface's features into a broadband signal rather than a well separated signal specific to each feature. Indeed, looking at optical satellite images, surface streams in Greenland appear as a wide blue line overlying clean ice, which is not the case for Svalbard glacier's streams. Moreover, in Svalbard, there is extensive ice foliation and a network of shear fractures caused by friction with the surrounding mountains that closely resemble surface streams.

The consistency of surface streams on a decadal timescale (Fig.5(a)), especially on the lower part of Werenskioldbreen, could be explained by the weak dynamics of the front and the fact that the longer a stream remains active and the deeper it carves into the glacier's surface, the more likely it is to survive. Despite these similarities, we observed several changes in the supraglacial drainage system on a decadal timescale in response to changes in geometry caused by climate warming and glacier flow. The occurrence of new moulins and new crevasses has a direct impact on the subglacial drainage system by creating new WIAs followed by potentially new subglacial channels. Abandoned moulins also deactivate the englacial and subglacial conduits they previously supplied with water (Holmlund, 1988; Nienow et al., 1998; Poinar et al., 2015). In the absence of internal water pressure and erosion, such conduits may then be closed by ice deformation. Nevertheless, observed supraglacial changes does not seem to imply a complete reorganization of the subglacial system (e.g. like after a surge event). In fact, new WIAs are either in a relatively close area (about 300 m$^2$) of the old ones or on the same subglacial channels axes than the pre-existing ones. This is especially the case with abandoned moulins, which are deactivated by new moulins a short distance up-glacier.

It is therefore crucial to investigate the permanency of the supraglacial drainage system of this glacier because it determines the validity of the duration of our model. Supraglacial drainage patterns are relatively persistent on an annual timescale (Fig.5(b)) as expected from the study of Nienow and Hubbard (2006). Their study suggests that a subglacial drainage system remains relatively steady from year to year. Finally, considering the low impact of decadal timescale changes in the supraglacial drainage system (Fig.5(a)) on the subglacial system, we can consider that our model is valid, with some minor changes, for a minimum period of 20 years. Results of the subglacial drainage system modelled for the year 1936 and the period 2005-2008 by Grabiec et al. (2017) reinforced our statement by displaying only a few changes regarding the subglacial patterns and the outflow locations.

The three different water pressure states of simulations described in section 4.2 may be representative of two different melt season periods:

- $K = 1$ and $K = 0.85$ may represent the beginning of the melt season when a considerable amount of water is delivered to the hydrological system due to melting of the winter snow mantle. It may also illustrate heavy rainfall and ice melt events. In these conditions, the channels are filled or nearly filled with water, $P_w = P_i$ or $P_w = 0.85 P_i$, and water flow may be mainly controlled by the surface topography of the glacier (Flowers and Clarke, 1999).

- $K < 0.85$ may represent all the other periods of the melt season. Especially after a high melting event ($K$ value will be close to 0) when the conduits have been extensively enlarged, caused by the walls melting due to the frictional heat released by an intense turbulent water flow, combined with a small water input volume. Under such conditions, the channels are not full of water, $P_w < P_i$ and the water flow may be controlled mainly by the bedrock topography of the glacier (Hagen et al., 2000; Sharp et al., 1993).

The total calculated volume of water entering Hansbreen over the whole 2015 melt season is 132.7 $10^6$ +/−4.2 $10^6$ m$^3$, of which 74% is meltwater and 26% precipitation. Taken together, the theoretical patterns of modelled subglacial channels indicate that most of the water is drained by a main conduit located on the east side of the glacier (Fig.6). Both scenarios exhibit more or less the same subglacial conduits in the ablation area as a result of a heavily crevassed surface. In fact, in scenario (2) most of the glacier surface in this area is considered as a WIA (Fig.2(a)) whereas in scenario (1) the entire

surface is considered as a WIA. Therefore, in scenario (2), only 47% of the total water input volume, of the ablation area, is drained in a discrete manner whereas the percentage for Werenskioldbreen is 100% (Fig.4). This is due to the fact that in general, tidewater glaciers are more crevassed than land-terminating glaciers because their greater dynamics are related to the difference in the morphology of their fronts (Larsen et al., 2007; Moon and Joughin, 2008; Van der Veen, 2007) whereas their subglacial hydrology system is very similar. However, scenario (1) displays some subglacial channels in the accumulation area

while scenario (2) does not. This is the main improvement represented by our results, in considering a discrete water recharge for this type of glacier. Scenario (2) in which $K = 0.85$ (Fig.6(e)) is most in agreement with field observations. In fact, it is the only scenario that represents the well-studied subglacial channels generated by Crystal Cave and Bird Brain Cave (Gulley et al., 2012; Murray et al., 2007; Schroeder, 1998; Turu, 2012), with a coherent orientation compared to existing maps (Benn et al., 2009; Mankoff et al., 2017) and the authors' personal observations, along with the best location of outflows. In fact, the

authors visited those two cave systems several times a year since a few years and repeated observations confirmed that the data cited above are still valid. However, the locations of the modeled outflows do not correspond perfectly with our observations and the one made by Grabiec et al. (2017); Pälli et al. (2003) (locations of sediment plumes, turbid water spots and visible R-channel). This is due to a lack of GPR data at the glacier front because of the presence of too many crevasses. The three outflows have their own water catchments and therefore drain different amounts of water (Fig.6(e)). According to our results,

the western outflow drained 2.7% of the total water volume, the central outflow 14.8% and the eastern one 82.5%.

The total calculated volume of water entering Werenskioldbreen over the whole 2015 melt season is 43.8 $10^6$ +/−1.4 $10^6$ m$^3$ of which 68.7% meltwater and 31.3% precipitation. The total annual runoff from the Werenskioldbreen basin from 2007 to 2012 measured by Majchrowska et al. (2015) ranged between 56.37 and 98.71 $10^6$ m$^3$. First, there is notable variability from

year to year. Second, the Werenskioldbreen basin considered in the study by Majchrowska et al. (2015) included the glacier forefield, meaning their study area was larger than ours. Third, we only considered the runoff from 06/06/2015 to 10/10/2015, not for the whole year. Finally, we underestimate the total runoff by only taking the water derived from precipitation and surface melting into account. For these reasons, we can be quite satisfied with our modeled value, which is of about the same order of

magnitude as those measured in the preceding years. Regarding our distribution of water sources, we can also be quite satisfied when we compare it with calculations made by Majchrowska et al. (2015). In 2009, total runoff comprised 71% meltwater, 17% precipitation and 9% other sources, and in 2011 it was 63% meltwater, 28% precipitation and 9% other sources. All the model results of the theoretical pattern of subglacial channels indicate that most of the water is drained by a main conduit located in the central part of the glacier whose outflow is located further south of the medial moraine (Fig.7). Both scenarios produce

different results because of the low glacier dynamics leading to a poorly crevassed surface resulting in considerable differences in water recharge between the spatial (Fig.3(b)) and the discrete (Fig.4(b)) test cases. Scenario (2) in which $K = 1$ (Fig.7(d)), with one outflow located in the northern part of the medial moraine and two outflows in its southern part, best matches field observations. In fact, the modeled locations of the outflows in this simulation fit quite good our observations and the one made by Grabiec et al. (2017); Majchrowska et al. (2015); Pälli et al. (2003) (location of streams in the glacier forefield). The three

outflows have their own water catchments and therefore drain different amounts of water (Fig.7(d)). The northern one drained 35% of the total water, the central one 51.6% and the southern one 13.4%. The fact that our model does not appear to be influenced by Werenskioldbreen medial moraine, unlike the model of Pälli et al. (2003), may be due to the new geometry of the glacier in 2015 compared to that in 1999. However, some dye tracing measurements have shown that subglacial water can flow across this medial moraine area during about the same modeling period as that used in the study by Pälli et al. (2003).

Regarding Werenskioldbreen's moulins, we observed that those located at higher elevations are supplied by more precipitation than meltwater, contrary to the moulins located at lower elevations e.g. two moulins situated at 420 m a.s.l. are supplied by 69% meltwater and 31% precipitation and one moulin situated at 112 m a.s.l. is supplied by 85.7% meltwater and 14.3% precipitation. Such observation shows that water recharge regarding melt and precipitation water proportion is heterogeneous on the glacier surface. This tendency was not observed on Hansbreen, probably due to its less steep slope and the smaller range

of elevation of the cold ice surface compared to Werenskioldbreen.

The fact that discrete recharge models generate subglacial channels starting at lower elevations and prevent any conduits from being generated in the accumulation area may be consistent with reality. In fact, we know that water does not penetrate the bare ice surface of a glacier without the presence of crevasses or moulins, (Fountain and Walder, 1998; Hodgkins, 1997; Lliboutry, 1971; Paterson, 1994; Ryser et al., 2013), and that water percolates through the snowpack and the firn to flow at the

temperate ice/firn interface, and also that WIAs are mainly located in the central part of the two glaciers we studied. In some simulations of Hansbreen with scenario (1) (Fig.6(a); 6(b) and appendix Fig.1(a)) we observed a subglacial channel in the accumulation area flowing outside the glacier in an eastern glacier system belonging to Paierlbreen. This could be an artefact due to the boundary of our model which does not allow Paierlbreen's drainage system to influence the Hansbreen drainage system, but not necessarily. In fact, since Paierlbreen surged in 2006, the ice division was drained downslope thereby reducing

Paierlbreen's ice pressure on Hansbreen and facilitating the flow of water from the upper part of the Hansbreen system to the Paierlbreen system than the contrary. Therefore, it may fit to the reality to observe this state under high water pressure conditions ($K = 1$, $K = 0.85$ and $K = 0.75$) (Fig.6(a); 6(b) and appendix Fig.1(a)) and not under atmospheric or low water pressure conditions ($K = 0$, $K = 0.25$ and $K = 0.5$) (Fig. 6(c), and appendix Fig.1(b); Fig.1(c)) because of the presence of a bedrock obstacle at this location. Also, a greater number of modeled channels are observed in the simulations in which $K = 1$, than in the simulation in which $K < 1$ (Fig.7). This may be due to the fact that when $K = 1$ new temporary subglacial channels are created due to higher water pressure in the distributed system (Hewitt, 2011). It is worth highlighting that we are able to observe these two phenomena even without the representation of the distributed drainage system in the model.

The volumes of water may be underestimated in the model. Indeed we did not take in account water stored in the snowpack and in the firn layer during the winter/spring period, which is then released during the melt season, nor subglacial meltwater produced at the glacier bed due to geothermal flux and melting of the subglacial channel walls due to the heat transfer induced by the water circulating within the conduits. Despite these simplifications in our estimations of water volumes, since surface meltwater is by far the most important source of water recharge in the subglacial system and that precipitation water is included in our model, we can assume that any water sources that are not taken into account in our study can be neglected Cogley et al. (2011); Hock (2005); Irvine-Fynn et al. (2011); Jansson et al. (2003).

## 6 Conclusions

The supply of water from surface melt is the most influential runoff component, confirmed by the difference of a factor of three in the amount of water provided by melt (72.5%) and precipitation (27.5%) during the 2015 melt season for Hansbreen and Werenskioldbreen.

We can conclude that changes in the supraglacial drainage system on a decadal timescale resulted in adjustments of the subglacial drainage system in response to the activation or deactivation of WIAs. Nevertheless, regarding our two study glaciers, the WIAs location undergo some change (about 300 m) but generally stay on the same subglacial axes which does not result in a fundamental reorganization of the subglacial system. On an annual timescale, the superficial drainage system of both glaciers remains spatially consistent, implying similar subglacial drainage systems.

The theoretical pattern of subglacial channels under water pressure conditions, ranging from atmospheric to ice overburden, was modeled taking into account local meltwater and precipitation and forcing water penetration inside the glacier thanks to identifying the locations of WIAs, for the melt season 2015.

It can be concluded that, considering discrete water recharge makes it impossible to form subglacial channels in most of the accumulation area formed by surface water supply which is consistent with previous theoretical studies (Fountain and Walder, 1998; Lliboutry, 1971). Concerning Svalbard tidewater glaciers, which have large crevassed areas, modeled patterns of theoretical subglacial channels assuming a spatial water recharge display some imprecisions but are far from being incorrect,

especially for the ablation zone. The same may be true for extensively crevassed glaciers during the active phase of a surge. On the contrary, it is important to consider a discrete water recharge for Svalbard land-terminating glaciers with limited crevassed areas (which is mainly the case in this type of glacier). This may be also true for long flat Svalbard tidewater glaciers or even glaciers in a quiescent phase of a surge. In any case, considering a discrete water recharge when modeling patterns of

theoretical subglacial channels makes it possible to achieve more realistic results.

     Changes in the location of subglacial channels depend to a great extent on changes at the surface (topography and supraglacial drainage system). The permanency of the supraglacial drainage system from year to year and the lack of major changes on a decadal timescale, allow us to consider our subglacial channels models valid, maybe with some slight changes, for a minimum period of 20 years.

This paper presents a new way of modeling the pattern of subglacial conduits of glaciers. It includes a discrete water recharge, based on a precise mapping of the entire glacier surface, and the volume of water runoff specific to all observed WIAs. Consequently, it produce more realistic results than was previously possible. Our model results are validated by observed locations of the outflows of subglacial channels at the front of our two studied cases. A more accurate reconstruction of the routes of subglacial water flow would require a model including englacial water transport and storage, drainage through a

subglacial water sheet (distributed drainage system) and subsurface groundwater flow. Our model also needs to be compared with a greater amount of field data such as dye tracing measurements and a survey of water discharge from several supraglacial streams sustaining moulins and of glacier outflows.

*Code availability.*   TEXT

*Data availability.*   TEXT

*Code and data availability.*   TEXT

*Author contributions.*   TEXT

     LD prepared structure of the paper and wrote the main part of the manuscript, created the DEMs of both glaciers, mapped the supraglacial system of both glaciers for the year 2015, determined the WIAs and calculated their catchments and the water input of the model. JJ and MG suggested subject of study and contributed to the paper design. MG also conducted the

subglacial water flow simulations. DI was responsible for the ablation and precipitation calculations and data acquisition. All authors contributed to interpretation of results and commenting, reviewing and editing the paper.

*Competing interests.* TEXT

The authors declare that they have no conflict of interest.

*Disclaimer.* TEXT

Appendix

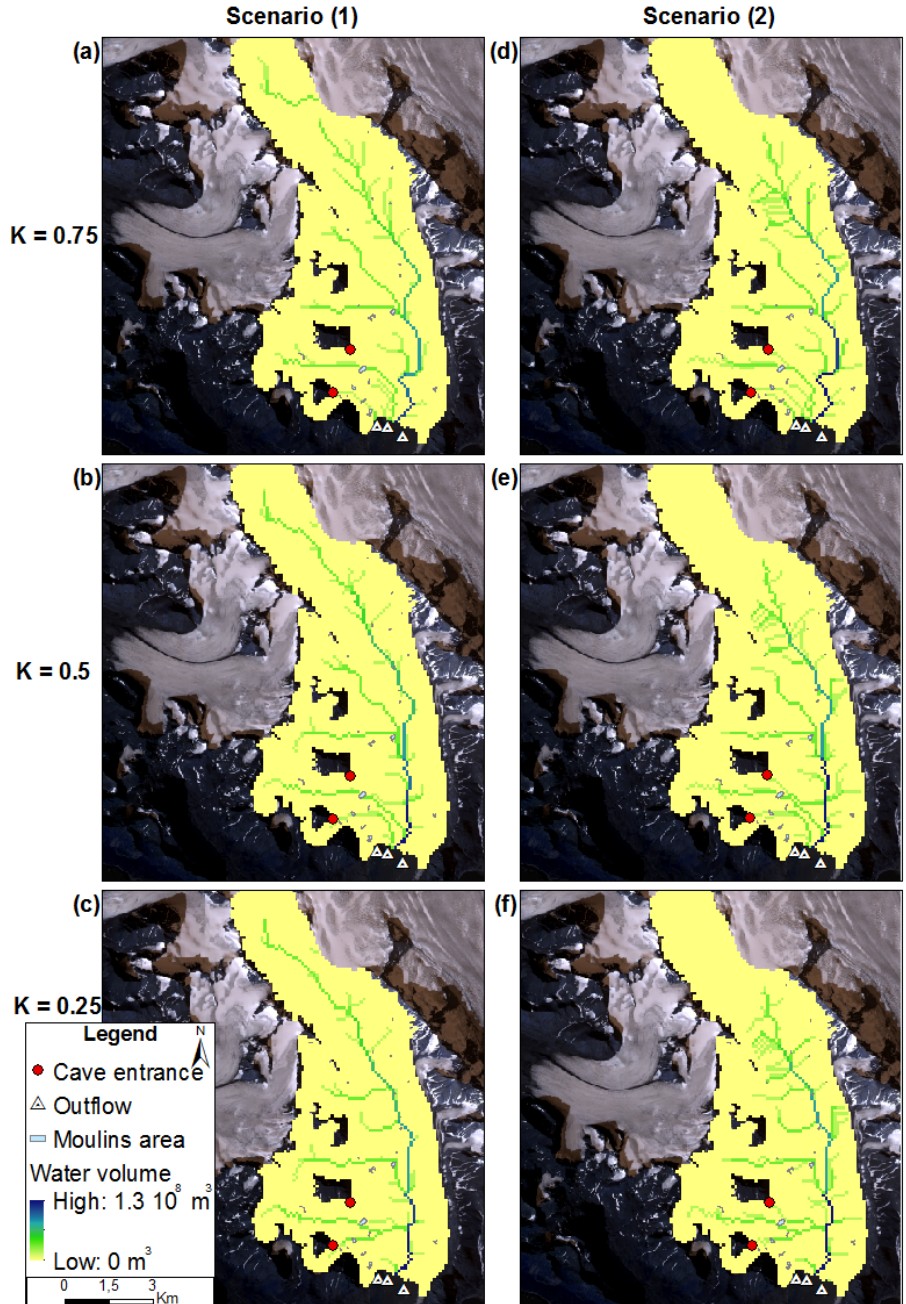

**Figure 1.** Map of the theoretical pattern of subglacial channels of Hansbreen modeled with scenario (1) ($K = 0.75$ (a); $K = 0.5$ (b); $K = 0.25$ (c)) and (2) ($K = 0.75$ (d); $K = 0.5$ (e); $K = 0.25$ (f)). The map background is a WorldView-2 VHRS image acquired on 21/08/2015.

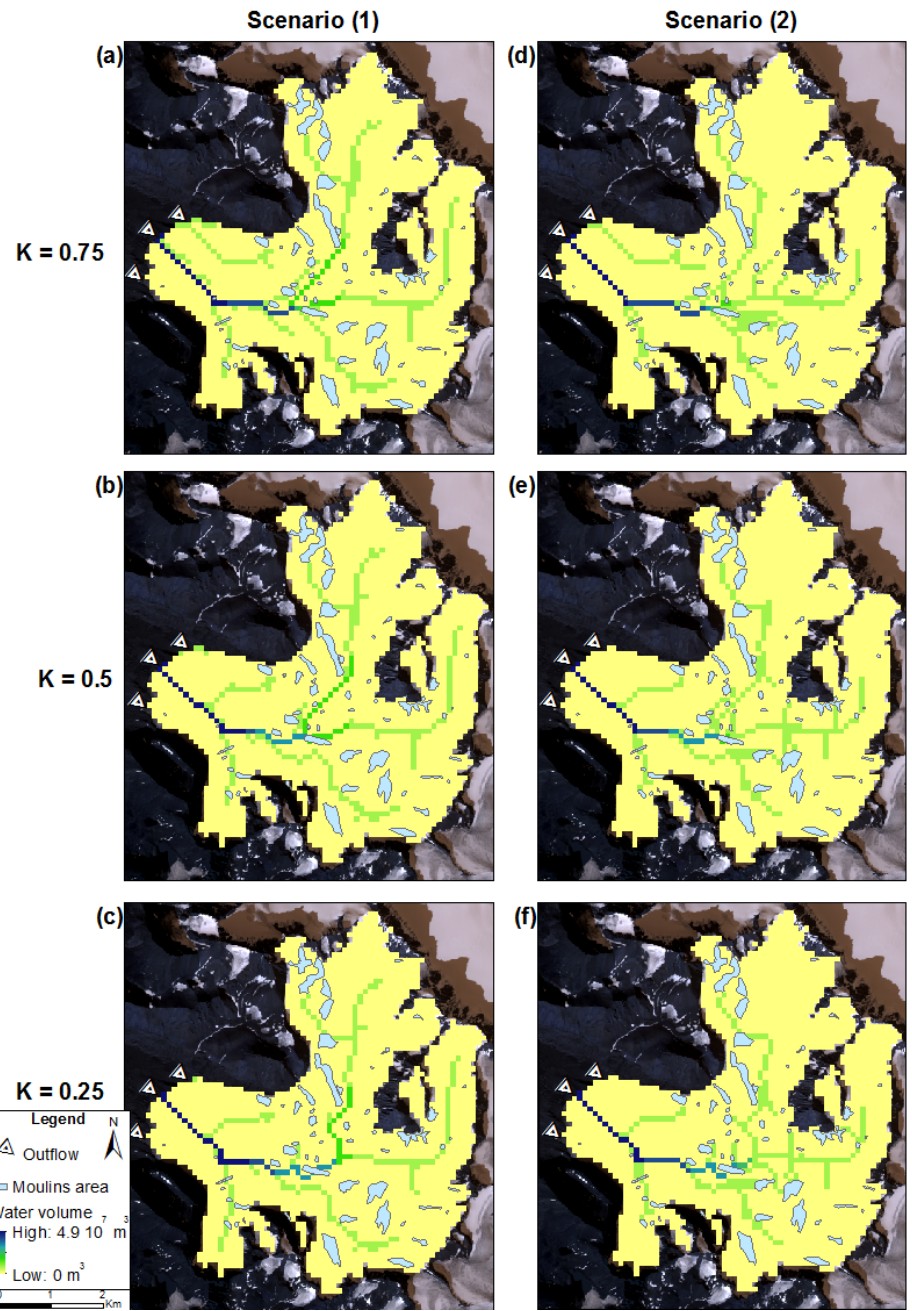

**Figure 2.** Map of the theoretical pattern of subglacial channels in Werenskioldbreen modeled with scenario (1) ($K = 0.75$ (a); $K = 0.5$ (b); $K = 0.25$ (c)) and (2) ($K = 0.75$ (d); $K = 0.5$ (e); $K = 0.25$ (f)). The map background is a WorldView-2 VHRS image acquired on 21/08/2015.

*Acknowledgements.* The authors would like to thank the European Space Agency for providing the WorldView-2 high resolution satellite images and SPOT images (Project no. C1P.34101 and no. C1P.9630). Fieldwork was supported by the Polish Ministry of Science and Higher Education (IPY/269/2006). Glaciological, hydrological and meteorological data were processed by the University of Silesia data repository within project Integrated Arctic Observing System (INTAROS). This project has received funding from the European Union's Horizon

5   2020 research and innovation programme under grant agreement No 727890. We wish to thank colleagues from the Polish Polar Station at Hornsund and Dariusz Puczko from the Institute of Geophysics Polish Academy of Sciences for hospitality and logistic support during field missions. Access to the meteorological data from the Hornsund station provided by the Institute of Geophysics, Polish Academy of Sciences is kindly acknowledged. The advanced stage of this work and preparation for publication was financed by the Centre for Polar Studies, University of Silesia - the Leading National Research Centre (KNOW) in Earth Sciences (2014-2018), No. 03/KNOW2/2014. We also thank

10  Doug Benn and two anonymous reviewers for reviews that improved the quality of the manuscript as well as the editor Olaf Eisen for his patience.

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
