# Peer review of "Role of discrete water recharge from supraglacial drainage systems in modeling patterns of subglacial conduits in Svalbard glaciers"

_The Cryosphere, 2017_

## Referee Comment (RC2) · Anonymous Referee #1 · 9 Mar 2018

The comment was uploaded in the form of a supplement:
https://www.the-cryosphere-discuss.net/tc-2017-219/tc-2017-219-RC2-supplement.pdf
* * *

---

## Referee Comment (RC3) · Anonymous Referee #1 · 14 Mar 2018

[a4paper,11pt]article [utf8x]inputenc [a4paper, margin=1.5cm]geometry

**Review of "Role of discrete recharge from the supraglacial drainage system for modelling of subglacial conduits pattern of Svalbard polythermal glaciers"**

**General comments**

This paper aims to assess the importance of supraglacial hydrology on subglacial water flow. I believe this to be a very important topic in the realm of subglacial hydrology, which could have important implications for other environments including Greenland. The paper presented here, makes use of classical formulations of hydraulic potential to define subglacial water flow paths, then by some mechanism, the model is run with discrete water inputs and a supraglacial hydrology model and the results are compared. It is possible that this simplistic approach could serve as a useful framework to quantify the effects of supraglacial hydrology on the subglacial drainage system. However, the steps that taken to apply the different hydrology distributions to the by means of the hydraulic potential in the subglacial system is not clear to me. Until this is clarified, it is not possible to judge the scientific quality of this work.

Many of the sentences, while mostly comprehensible, are difficult to follow. Some

terminology is misleading, confusing or incorrect. The presentation of some data is very poor and not of the standard of scientific publications (significant figures...) Lastly, the manuscript is poorly organized.

It appears that Mr. Decaux is a PhD student. I believe that his advisors must do a more thorough job of preparing this manuscript and guiding Mr. Decaux; it is not the job of reviewers to do this. Should the methods prove to be viable, I believe that the results, when fully described and presented in a logical manner, presented here could be a demonstration of variability in subglacial water flow caused by the different treatments of supraglacial hydrology. The results of the paper could be interesting, however, because of the issues with organization and the lack of clarity in the methods, I do not see how this can be accepted for publication. I hope to have provided some examples and comments, which when applied to the remainder of the manuscript, can improve it in the event of another submission.

**Terminology**

- **recharge**: This either needs a new word or to be specifically defined. Do you mean seasonal water flux? Would mean seasonal water discharge be better? Discharge sum?

- **discrete recharge**/ **homogeneous recharge**: I think this is misleading. Homogeneous recharge, to me, seems like putting water equally over the glacier bed. I might suggest the terms with/without supraglacial hydrology. Whatever the terminology, it must be made clear in the introduction/methodology. Sometimes the work **theory** is used, I think a better word would be "test case" or "scenerio"

- **precipitations**: This is not a word. Use precipitation.

- A large point is made of these glaciers being **polythermal**. However, I found it

hard to follow how this fit into the scheme presented here. Discuss the implications of it being a polythermal glacier, including in **Methods** and **Discussion** or only briefly mention in **Study area**.

- **conducts**: I assume you mean "conduit".

**Organization**

- **Abstract** This needs to be reworked. For improvement, I encourage you to look at resources such as this one: http://sepwww.stanford.edu/sep/prof/abscrut.html

- **Introduction** consists of three parts. 1) The background knowledge, broad to specific. 2) The gap in knowledge that the paper will address. 3) Your solution to the knowledge gap. Your work should only be discussed in the last paragraph or two, and should summarize what you will do. The description of your work that takes up most of the introduction here does not belong here.

- **Page 5 Section 3.1.1.** This paragraph is way to long. Try making it just a couple of sentences describing the overall scheme. Put dates, datasets, etc. in the respective subsections. Also make this section 3.1. Consider changing the name of the previous section to **Study sites and datasets** and put the dataset description there. i.e. weather data, imagery.

- I would encourage you to make the **descriptions of experiments** a separate subsection. Its title could be "model runs" or "experiment design".

- **Page 10 Lines 6-8** You present several different K values. However, your results only utilize a few. Either exclude the others, or make the plots/maps available in some kind of supplementary material.
- I found quite a few **Results** presented in the **Discussion** section. This is an big issue with this paper. Separating these topics can be difficult. This needs to be reorganized and consider if combining the sections could be a good idea.

- **Supraglacial drainage system evolution**. This section has some interesting little facts, however, I don't see how it is coherent to the overall story of the paper (and I am not entirely sure what the greater story is). Either find a way to make it relevant, or excluded it.

- **Page 13 Lines 1-12** These descriptions of K should be condensed and added to methods.

- **Page 13 Lines 13-15** Where are these caves discussed? How are caves **better represented with a discrete water recharge**? This needs to be clearer

- **Page 17 Lines 10** Discuss why this is important to your work.

- **Page 17 Lines 12-13, Lines 27-29** I believe most people would believe amounts of water to be a result as opposed to something to be discussed. Also what to the $+$ and - stand for? What are the uncertainties? Standard deviations? Model errors? Variability in the input data? You present the water discharge quantity in confidence to the $m^3$. That seems like an overly precise measurement to me. This is not how numbers in are represented in science, look into significant figures .

- **Page 17-18** You repeatedly talk about outflow observations. What kind of data is this? Another paper (then cite)? if not, you need to make section in your **Methods/Study area** to describe how this data was collected. This shows that your model can somehow be verified (a very good thing. . . ). Making a point of describing this will strengthen your conclusion.

- **Page 18 Lines 26** underestimated not under-estimating.

- **Page 20** I think you could probably end your paper on a stronger note by high-lighting the difference between the two situations.

**Scientific Comments**

- **Page 2 Lines 1-6** Why is subglacial hydrology important? Sliding? subglacial sediment transport? ice marginal lake drainages?

- Form **Page 2 Line 7 to Page 3 Line 10** This needs to be rewritten, omitting what is done in your experiment, and using the formulation above.

- **Page 2 Lines 33** Many have worked on the role on non-uniform water input on subglacial hydrology, these include Werder et. al. (2013), Hewitt (2013), Poinar et al., (2015).

- **Methodology, Spatialized water runoff calculation** Shorten to "Spatially distributed runoff estimation". More importantly, it is not clear how this is done. Do you use a T-index model? Do you prescribe the discharge with an elevational gradient/hypsometry? This needs a better explanation.

- **Page 8 Line 17**. Where does the bed DEM come form?

- **Methodology** You need a section detailing how the $Q_h$ is coupled to the sub-glacial model. The whole paper relies on this, and it is not evident to me what was done. This is one reason why I don't find the paper to be publishable. There might be some hints in the list of model scenarios, but I am not sure. I believe this coupling should have its own subsection.

- **Page 10 Lines 1-2** No. Röthlisberger 1972 and Weertman 1972, discuss englacial water flow, not surface water flow. Also this paragraph moves around

quite a bit. I recommend finding a linear line of reasoning for your choices in K and then present that.

- **Page 17 Lines 7-9** There has been plenty of work examining moulins, if you choose to discuss them, then add citations.

- **Page 18 Lines 12-14** How do you know that water does not penetrate the cold glacier ice? Do you have field observations? or did you read about this in a paper (which you then NEED to cite)?

- **Page 19 Lines 20-23** How does subglacial hydrology differ between tidewater and land-terminating glaciers? How do tidewater glaciers become more crevassed? This sections need more citations and should go in the discussion section. Poinar et al., (2015) discusses the implications of surface hydrology and moulins. It would be nice to put this work in relation to that research.

- **Page 19 Lines 24-28** Here it seems like you are trying to discuss the future of the subglacial drainage system, with the moving moulins due to the flow of ice. However, as Fischer et al. (2005) point out surface elevation changes (do to increased melt/glacier retreat) also will affect the location of the subglacial drainage system. This is another point worth mentioning looking toward the future.

**Figures**

- I do not think that the coordinate system used in necessary to mention.

- **Figure 1** Mass balance, not Masse balance. Also adding some interpreted flowlines will help the reader orient themselves.

- **Figures 7 and 8** What are the different columns? Mention this in the caption. Also I find the UTM markers a bit distracting and not necessary. These should be removed.

---

## Referee Comment (RC4) · D. Benn (Referee) · 14 Mar 2018

This paper looks at the relationship between surface water inputs and subglacial drainage networks, an important topic that previously has received relatively little attention. I especially like the careful mapping of surface drainage networks and the identification of moulins and other water input points, and the use of these data as inputs for the subglacial water routing model. The paper thus has the potential to be a very useful addition to the literature.

[Figure]

However, the paper needs a lot of work before it is ready for publication. Referee 1 has made specific recommendations for tightening up the paper structure, with which I agree. These recommendations will help to create a clearer logical progression from observations to analysis, followed by validation of results and interpretation. This will clarify the aims, results and conclusions of the paper, increasing its clarity and impact. I also agree with Referee 1 that the language requires thorough checking throughout. The grammar, spelling and sentence structure all need careful revision, ideally with the help of a native English speaker. I don't know how 'conduit' came to be spelled 'conduct' throughout the paper, but the fact that this fundamental term came to be misspelled in a paper about glacier hydrology highlights the extent of the problem.

For the most part, the methods are sound and the assumptions reasonable. One of the assumptions, however, is highly questionable with some implications for the reliability of the results. On p. 4, line 3 and following, it is stated: "In the accumulation area temperate ice and firn are present, allowing water percolation through the glacier body..." Using this assumption, the entire upper part of Hansbreen is defined as a WIA (p. 6, line 23; p. 9, line 28). Temperate ice was considered to be permeable by some early authors (e.g. Nye and Frank, 1973), but this is no longer the accepted view (see Fountain and Walder, 1998). The paper by Lliboutry (1971: J Glac 10, 15-29) is well worth reading on this topic. He provides detailed observational evidence and theoretical considerations that show that bulk temperate ice cannot be permeable. These considerations thus invalidate the assumption that water can directly access the bed wherever temperate ice occurs through the whole glacier thickness (such as upper Hansbreen). Instead, firn aquifers are perched above essentially impermeable ice, with the transition occurring at about 30 m depth in the European Alps. Thus inclusion of areas of temperate ice as WIAs (Fig. 3a) is thus not justified. The WIA on Hansbreen should be redrawn omitting the temperate ice zone, and the model re-run.

Independent validation of the model results is of course difficult, given that most of the drainage system below the glaciers remains unobserved. So the location of outflow

points (portals and plumes) is crucial. At present, this important information is not prominent enough in the text. It should be clearly flagged up as the key test of model output, ideally in a separate subsection labelled 'Model Validation'. The location of meltwater portals and upwellings should be plotted on Figs 7 & 8, so the reader can clearly compare the predicted conduit locations with known efflux points. It is also worth noting that a similar model to the one used by the authors was successful at predicting the location of plumes in front of the tidewater glacier Kronebreen (How et al., 2017: The Cryosphere), and also indicates that the K value varies through the melt season. The fact that Scenario (5), K = 0.85 represents subglacial channels observed at Crystal and Bird Brain Caves cannot be regarded as model validation. The modelled channels appear because the caves are specified as water influx points. The existence of subglacial channels extending from moulins at these locations is certainly worth highlighting, but this ought to be early in the paper - perhaps in Section 3.2 where the observational data are introduced.

Figures 7 & 8 are very interesting, but their impact can be greatly increased by simply changing the arrangement of the panels. At present, it is very difficult to assess the result and involves much tiresome switching back and forth between caption and panels. Instead, the panels should be arranged so that the two columns show Scenarios 2 & 5, and the three rows show K = 1, 0.85 & 0. This will immediately allow readers to see how the water input and pressure assumptions influence the results.

On p. 8, line 9: the possibility is mentioned that additional water might be released from winter/spring storage in the snowpack. What about the opposite possibility? i.e. how much of the calculated melt might be retained in snow/firn? Does the model simply assume that all meltwater will enter supraglacial/subglacial transport, with zero surface storage? This issue is related to the erroneous attribution of the whole accumulation area of Hansbreen as a WIA. If meltwater in this zone is retained in the snowpack, instead of being immediately transferred to the bed, then this will significantly reduce the modelled water inputs to the bed possibly with major implications for the results.

[Figure]

In summary, this is an interesting paper with a lot of potential. I look forward to seeing a revised version that maximises the impact of the results.

---

## Editor Comment (EC1) · O Eisen (Editor) · 16 May 2018

Dear Léo Decaux,

the two referees provided an extensive evaluation of your manuscript and identified substantial weaknesses. I therefore ask you to consider all of them thoroughly in the revised version of your ms. Please take special care for those issues related to assumptions and methodology. Based on the revision I will evaluate whether the ms is in principle suitable for publication in TC and a further round of reviews.

[Figure]

Regards, Olaf

Co-editor in Chief for TC

---

## Author Response (AR1)

**Review of \Role of discrete water recharge from supraglacial drainage systems in modeling patterns of subglacial conduits in Arctic glaciers"**

All the points of the review below were taken into account during the review.  The title was changed; the Abstract and the Introduction were completely rewritten; all the other parts were reworked; figure 2 was deleted; appendix part was created; all the English language of the text was reworked. Basically, the entire article was modified.

Referee 1.

**General comment**

- Comment from Referee:

This paper aims to assess the importance of supraglacial hydrology on subglacial water flow. I believe this to be a very important topic in the realm of subglacial hydrology, which could have important implications for other environments including Greenland. The paper presented here, makes use of classical formulations of hydraulic potential to define subglacial water flow paths, then by some mechanism, the model is run with discrete water inputs and a supraglacial hydrology model and the results are compared. It is possible that this simplistic approach could serve as a useful framework to quantify the effects of supraglacial hydrology on the subglacial drainage system. **However, the steps that taken to apply the different hydrology distributions to the subglacial system is not clear to me. Until this is clarified, it is not possible to judge the scientific quality of this work.**

Author's response:

All the grid cells of the subglacial model are weighted by the total amount of water penetrating the glacier or not thanks to our supraglacial drainage mapping and the modelling of the different water catchment areas.

Author's changes in manuscript:

Few sentences will be add to clarify the methodology.

- Comment from Referee:

Many of the sentences, while mostly comprehensible, are difficult to follow. Some terminology is misleading, confusing or incorrect. The presentation of some data is very poor and not of the standard of scientific publications (significant figures…) Lastly, the manuscript is poorly organized.

It appears that Mr. Decaux is a PhD student. I believe that his advisors must do a more thorough job of preparing this manuscript and guiding Mr. Decaux; it is not the job of reviewers to do this. Should the methods prove to be viable, I believe that the results, when fully described and presented in a logical manner, presented here could be a demonstration of variability caused by the different treatments of supraglacial hydrology. The results of the paper could be interesting, however, because of the issues with organization and the lack of clarity in the methods, I do not see how this can be accepted for publication. I hope to have provided some examples and comment, which when applied to the remainder of the manuscript, can improve it in the event of another submission.

Author's response:

I totally agree for the language mistakes, that is why, as we have no native English involve in the paper, it will be send to a professional person to correct the language before the final submission.
Even if the mapping of the supraglacial system was done manually, we believe that the combination of high resolution satellite images with field observations and GPS measurements gave us a good representation of the supraglacial system of the two glaciers.

Author's changes in manuscript:

English language will be corrected.
The abstract and the introduction will be reworked as mentioned below. Also some part of the subsection will be move between each other as mention below in the view of a better organization of the paper which will make the paper clearer.

**Terminology**

- Comment from Referee:
  **Recharge**: This either needs a new word or to be specifically defined. Do you mean seasonal water flux? Would mean seasonal water discharge be better? Discharge sum?

  Author's response:

  **Recharge** is a term widely used in the literature, e.g.:

Benn, Douglas, et al. "Englacial drainage systems formed by hydrologically driven crevasse propagation." Journal of Glaciology 55.191 (2009): 513-523.

Siegel, D. I., and R. J. Mandle. "Isotopic evidence for glacial meltwater recharge to the Cambrian-Ordovician aquifer, north-central United States." Quaternary Research 22.3 (1984): 328-335.

Ofterdinger, U. S., et al. "Environmental isotopes as indicators for ground water recharge to fractured granite." Ground water 42.6/7 (2004): 868.

Ofterdinger, U. S., et al. "Environmental isotopes as indicators for ground water recharge to fractured granite." Ground water 42.6/7 (2004): 868.

Döll, Petra, and Kristina Fiedler. "Global-scale modeling of groundwater recharge." Hydrology and Earth System Sciences Discussions 4.6 (2007): 4069-4124.

Gulley, J. D., et al. "The effect of discrete recharge by moulins and heterogeneity in flow-path efficiency at glacier beds on subglacial hydrology." Journal of Glaciology 58.211 (2012): 926-940.

**Recharge** refers to any input of water in the system (occurring mainly during the ablation season).

Author's changes in manuscript:

**Recharge** will be better defined in the abstract and in the Introduction.

- Comment from Referee:

**discrete recharge/ homogeneous recharge**: I think this is misleading. Homogeneous recharge, to me, seems like putting water equally over the glacier bed. I might suggest the terms with/without supraglacial hydrology. Whatever the terminology, it must be made clear in the introduction/methodology. Sometimes the work theory is used, I think a better word would be "test case" or \scenario"

Author's response:

Yes you well understood the notion of Homogeneous recharge, it is "like putting water equally over the glacier bed" or having water put all over the glacier bed in function of the spatialized water runoff model used (like explain page 9 with the different simulation scenarios). It will be clearer to use "spatially uniform recharge" instead of "homogeneous recharge" as used by Gulley et al 2012

Author's changes in manuscript:

"Homogeneous recharge" will be change into "spatially uniform recharge".

"discrete recharge" will be better defined in the abstract and introduction part.

We will replace "homogeneous water recharge theory" by "spatially uniform recharge test case".

We will replace "discrete water recharge theory" by "discrete water recharge test case".

We will change "hydraulic potential theory" into "hydraulic potential gradient" or "hydraulic potential surface".

- Comment from Referee:

  **precipitations**: This is not a word. Use precipitation.

  Author's response:

  You are right.

  Author's changes in manuscript:

  We will replace "precipitations" by "precipitation"

- Comment from Referee:

  A large point is made of these glaciers being **polythermal**. However, I found it hard to follow how this fit into the scheme presented here. Discuss the implications of it being a polythermal glacier, including in **Methods** and **Discussion** or only briefly mention in Study area.

  Author's response:

  We made a point on the fact that they are polythermal because considered cold ice as impermeable and warm ice as permeable. But as mentioned by the second referee, warm ice is not considered as permeable anymore. It will not change our results and conclusions but it will allows us to increase the impact of our article by extending our results for all glaciers (polythermal, temperate and cold) displaying a channelized supraglacial system with moulins and/or crevassed areas. So we will focus on the different surfaces properties:
  - firn = infiltration and englacial circulation.
  - bare ice (cold or warm) displaying stream on the surface = runoff.
  - crevassed area = penetration of water.

  Author's changes in manuscript:

We will correct this point in the different parts of the article.

- Comment from Referee:

**conducts**: I assume you mean \conduit".

Author's response:

You are right.

Author's changes in manuscript:

We will replace "conducts" by "conduits".

**Organization**

- Comment from Referee:

  **Abstract** This needs to be reworked. For improvement, I encourage you to look at resources such as this one: http://sepwww.stanford.edu/sep/prof/abscrut.h

  Author's response:

  You are right.

  Author's changes in manuscript:

  The abstract will be improve.

- Comment from Referee:

  **Introduction** consists of three parts. 1) The background knowledge, broad to specific. 2) The gap in knowledge that the paper will address. 3) Your solution to the knowledge gap. Your work should only be discussed in the last paragraph or two, and should summarize what you will do. The description that takes up most of the introduction here does not belong here.

  Author's response:

  You are right.

  Author's changes in manuscript:

  We will develop more the background knowledge about the models already existing and better precise the gap in knowledge. We will better summarize our work and focus more on how we will answer to the gap mentioned at the beginning of the introduction. Also we will move a part of the actual introduction in the methodology part.

- Comment from Referee:

  **Page 5 Section 3.1.1.** This paragraph is way too long. Try making it just a couple of sentences describing the overall scheme. Put dates, datasets, etc. in the respective subsections. Also make this section 3.1. Consider changing the name of the previous section to **Study sites and datasets** and put the dataset description there. i.e. weather data, imagery.

  Author's response:

  You are right.

  Author's changes in manuscript:

  We will put this paragraph into study sites and datasets. We will make this paragraph shorter by removing all the explanations on the reason of the use of VHRS.

- Comment from Referee:

  I would encourage you to make the **descriptions of experiments** a separate subsection. Its title could be "model runs" or "experiment design".

  Author's response:

  You are right.

  Author's changes in manuscript:

  We will make an extra subsection for the description of the different scenarios calls "model runs".

- Comment from Referee:

  **Page 10 Lines 6-8** You present several different K values. However, your results only utilize a few. Either exclude the others, or make the plots/maps available in some kind of supplementary material.

  Author's response:

  We explain why we do not present the results for some K values page 13 lines 1-3.
  The fact that the whole K values are mentioned allows us to justify why we are focusing on the three K value K=0; K=0.85 and K=1. Therefore it seems essentials to mention them to show the logic of our selection.

  Author's changes in manuscript:
  We make available 6 extra maps per glacier with discrete and homogeneous recharge for K=0.75; K=0.5 and K=0.25 in an appendix part.

- Comment from Referee:

I found quite a few **Results** presented in the **Discussion** section. This is a big issue with this paper. Separating these topics can be difficult. This needs to be reorganized, and consider if combining the sections could be a good idea.

Author's response:

We do not agree with this statement. Here is the list of the results presented compare to the discussion part. One thing should move from Results part to Discussion part, it is the interpretation of different K scenarios.
Also we prefer to keep the classical way of an article organization without combining the two sections.

Results:
- Supraglacial map description
- Interpretation of different K scenarios
- Hansbreen simulation description + no water volume but identification of main channel
- Werenskioldbreen simulation description + no water volume but identification of main channel

Discussion:
- Supraglacial map discussion
- Discussion of Hansbreen simulation + water volume
- discussion of Werenskioldbreen simulation + water volume
- Discrete recharge/Homogeneous recharge
- Connection with Paierlbreen
- Limit of the model in water volume values

Author's changes in manuscript:

We will move from Results part to Discussion part the interpretation of different K scenarios.

- Comment from Referee:

**Supraglacial drainage system evolution**. This section has some interesting little facts, however, I don't see how it is coherent to the overall story of the paper (and I am not entirely sure what the greater story is). Either find a way to make it relevant, or excluded it.

Author's response:

It is important because it allows us to estimate the validity duration of our model.
This timescale evolution might be the same for all the Arctic glaciers and because we demonstrate that it plays a very important role on the location of the subglacial channels, it is crucial to analyze its evolution through the time.

Author's changes in manuscript:

We will develop a bit more in the discussion part the fact that it allows us to estimate the validity duration of our model and why it is important.

- Comment from Referee:

**Page 13 Lines 1-12** These descriptions of K should be condensed and added to methods.

Author's response:

We agree that we have to move it but it should be in the discussion part because we find this classification:
- K=1; K=0.85 (because both of them display more than one outflow)
- K=0.75; K=0.5; K=0.25; K=0 (because all of them display one outflow)
thanks to our results. Therefore, we can not move it in to methods.

Author's changes in manuscript:

We will move from Results part to Discussion part the interpretation of different K scenarios.

- Comment from Referee:

**Page 13 Lines 13-15** Where are these caves discussed? How are caves better represented with a discrete water recharge? This needs to be clearer

Author's response:

The caves are only mark on the map in study area section. You are right, it is necessary to mentioned them in the text in this section. Also, we will explain how it is better represented with K=0.85 discrete recharge scenario (one or two sentence).

Author's changes in manuscript:

We will mention the caves in the text of study area section. We will explain how it is better represented with K=0.85 discrete recharge scenario (one or two sentence).

- Comment from Referee:

**Page 17 Lines 10** Discuss why this is important to your work.

Author's response:

You are right it is developed in the conclusion page 19 lines 7-9 but not in the discussion part.

Author's changes in manuscript:

We will develop a bit more on how the supraglacial system evolution tells about the subglacial system evolution.

- Comment from Referee:

**Page 17 Lines 12-13, Lines 27-29** I believe most people would believe amounts of water to be a result as opposed to something to be discussed. Also what to the + and - stand for? What are the uncertainties? Standard deviations? Model errors? Variability in the input data? You present the water discharge quantity in confidence to the m$^3$. That seems like an overly precise measurement to me. This is not how numbers in are represented in science, look into significant figures .

Author's response:

We can oppose our total water input volume calculated for 2015 with the total runoff of all the catchment of Werenskioldbreen area measured on the field for the years 2007-2011 by Majchrowska et al., 2015. This is not possible for Hansbreen as it is a tidewater glacier. Also we can compare the data of having 68.7% of the water supply by meltwater with observations made by Majchrowska et al., 2015.  With those comparisons we know that we achieve realistic values. +/- is the calculated error of our spatialized water runoff calculation.
You are right the values should not be present with a confidence to the m$^3$.

Author's changes in manuscript:

We will validate our spatialized water runoff calculation by comparing Werenskioldbreen values with Majchrowska et al., 2015 values.
We will quickly explain our calculation error.
We will change our m$^3$ values confidence. For example water values will be presented with this confidence 43.81 10$^6$ m$^3$.

- Comment from Referee:

**Page 17-18** You repeatedly talk about outflow observations. What kind of data is this? Another paper (then cite)? if not, you need to make section in your **Methods/Study area** to describe how this data was collected. This shows that your model can somehow be verified (a very good thing…). Making a point of describing this will strengthen your conclusion.

Author's response:

You are right. There were some outflows positions mapped in the past and it fit quite well with our model.

Author's changes in manuscript:

We will cite Palli et al 2003, it fit perfectly with the outflows map which confirm the validity of the model of about 20 years. Also fit in Grabiec 2017 for Hansbreen. It fit also with Werenskioldbreen outflows publish in Majchrowska, 2015. Finally, it fit with personal field observations from the authors.
We will also add some outflow position mapped on our two results figures.

- Comment from Referee:

**Page 18 Lines 26** underestimated not under-estimating.

Author's response:

You are right.

Author's changes in manuscript:

We will replace "under-estimating" by "underestimated".

- Comment from Referee:

  **Page 20** I think you could probably end your paper on a stronger note by highlighting the difference between the two situations.

  Author's response:

  This is done page 19 lines 19-23.

  Author's changes in manuscript:

  We will add a sentence, page 19, saying that in any cases, considering a discrete recharge display more accurate results.

**Scientific Comment**

- Comment from Referee:

  **Page 2 Lines 1-6** Why is subglacial hydrology important? Sliding? subglacial sediment transport? ice marginal lake drainages?

  Author's response:

  You are right, we mention the impact on dynamic in the abstract but not in introduction: part of the work which should be done on the introduction as mentioned before.

  Author's changes in manuscript:

  We will improve the introduction.

- Comment from Referee:

  **Form Page 2 Line 7 to Page 3 Line 10** This needs to be rewritten, omitting what is done in your experiment, and using the formulation above.

Author's response:

You are right: part of the work which should be done on the introduction as mentioned before.

Author's changes in manuscript:

We will improve the introduction.

- Comment from Referee:

  **Page 2 Lines 33** Many have worked on the role on non-uniform water input to hydraulic models, these include Werder et. al. 2013, Hewitt 2013, Poinar et al., 2015.

  Author's response:

  You are right: part of the work which should be done on the introduction as mentioned before.

  Author's changes in manuscript:

  We will improve the introduction.

- Comment from Referee:

  **Methodology**, **Spatialized water runoff calculation** Shorten to "Spatially distributed runoff estimation". More importantly, it is not clear how this is done. Do you use a T-index model? Do you prescribe the discharge with an elevational gradient/hypsometry? This needs a better explanation.

  Author's response:

  You are right.

  Author's changes in manuscript:

  We will rename the section into "Spatially distributed runoff estimation".
  We will better describe the relation between summer mass balance and elevation mentioned in this section.

- Comment from Referee:

  **Page 8 Line 17**. Where does the bed DEM come from?

  Author's response:

  The origin of the DEM of the bedrock is well described in the dataset part with in addition the reference to Grabiec et al 2012.

  Author's changes in manuscript:

  Nothing.

- Comment from Referee:

  **Methodology** You need a section detailing how the $Q_h$ is coupled to the subglacial model. The whole paper relies on this, and it is not evident to me what was done. This is one reason why I don't find the paper to be publishable.

  Author's response:

  You are right, the way how the water input data are apply to the model is explained in the description of the scenarios, page 9 lines 8-26, but for some precise case not with a general point of view.

  Author's changes in manuscript:

  We will better explain the fact that we take into account all kind of precipitation.
  We will change Qh into something else to precise that is precipitation and not the total amount of water.
  We will precise that we obtain three input files with water values:
  - Precipitation
  - Ablation
  - Precipitation + ablation

  We will better develop the sentence page 8 line 23-24 by explaining how the water input data are apply to the model in a general point of view.

- Comment from Referee:

  **Page 10 Lines 1-2** No. Rothlisberger 1972 and Weertman 1972, discuss englacial water flow, not surface water flow.

  Author's response:

  You are right for Weertman.
  Regarding Rothlisberger, he says that the pressure in the conduits depend directly on the discharge so by the logic of the mechanisms involve, it implies that it depends on the recharge.

  Author's changes in manuscript:

  We will find other article to cite.
  We will keep Rothlisberger and explain better the relation.

- Comment from Referee:

  **Page 17 Lines 7-9** There has been plenty of work examining moulins, if you choose to discuss them, then add citations.

Author's response:

You are right.

Author's changes in manuscript:

We will add citations.

- Comment from Referee:

**Page 18 Lines 12-14** How do you know that water does not penetrate the cold glacier ice? Do you have field observations? or did you read about this in a paper (which you then NEED to cite)?

Author's response:

This is well known in glaciology.

Author's changes in manuscript:

We will add citations the first time we mention this phenomenon in introduction page 3 line 28 plus say that it is impermeable. (Paterson, 1994; Hodgkins, 1997; Ryser, C., et al, 2013).

- Comment from Referee:

**Page 19 Lines 20-23** How does subglacial hydrology differ between tidewater and land-terminating glaciers? How do tidewater glaciers become more crevassed? This sections need more citations and should go in the discussion section. Poinar et al., 2015 discusses the implications of surface hydrology and moulins. I would be nice to put this work in relation to that research.

Author's response:

You are right, it needs more precision. The fact that tidewater glaciers are more crevassed is well known in glaciology. This is due to the fact that in general, tidewater glaciers have a higher dynamic than land-terminating glaciers. There is no big differences between tidewater and land-terminating glacier's subglacial hydrology system, except due to the fact that tidewater glaciers are more crevassed.

Author's changes in manuscript:

In the discussion part we will develop the differences between tidewater and land-terminating glaciers and add new citations.
We will include Poinar et al 2015 and Van Der Veen 2007 citation.

- Comment from Referee:

  **Page 19 Lines 24-28** Here it seems like you are trying to discuss the future of the subglacial drainage system, with the moving moulins due to the flow of ice. However, as Fischer et al. 2005 point out surface elevation changes (do to increased melt, etc.) also will affect the location of the subglacial drainage system. This is another point worth mentioning looking toward the future.

  Author's response:

  The fact that we based our estimation of the subglacial drainage in the future on the evolution of the supraglacial drainage system which is the result of the glacier flow and the surface elevation changes, by definition, take into account those two components. In Fisher et al 2005, the impact of the glacier geometry changes on the subglacial system are noticeable mainly due to the presence of a medial moraine which is increasing and impact the system. This is not the case for our glaciers. Even if there is a medial moraine on Werenskioldbreen, it does not impact the subglacial system as the channels are passing bellow this medial moraine (observed by Czech explorations cf their field report).

  Author's changes in manuscript:

  Nothing.

**Figures**

- Comment from Referee:

  I do not think that the coordinate system used in necessary to mention.

  Author's response:

  It was not mentioned before but it was a request of the editor before the submission.

  Author's changes in manuscript:

  We will mention it for the first figure but not for the following ones knowing that it is the same.

- Comment from Referee:

  **Figure 1** Mass balance, not Masse balance. Also adding some interpreted flowlines will help the reader orient themselves.

Author's response:

You are right for mass balance. Concerning the flow direction, having the front part of the glacier visible + the topographic lines, in our point of view, it is not necessary to add some interpreted flowlines having the basic knowledge that a glacier is flowing from higher to lower elevation due to the gravity.

Author's changes in manuscript:

We will change "masse balance" for "mass balance".

- Comment from Referee:

  **Figures 7 and 8** What are the different columns? Mention this in the caption. Also I find the UTM markers a bit distracting and not necessary. These should be removed.

  Author's response:

  You are right for the UTM coordinate. There is nothing specific to the columns. It is just needed to refer to the legend and to the letter corresponding to the figure (a); b); etc…).

  Author's changes in manuscript:

  We will remove the UTM coordinates.
  Other changes will be applied in the organization of those two figures, cf answer to the other review bellow.

This paper looks at the relationship between surface water inputs and subglacial drainage networks, an important topic that previously has received relatively little attention. I especially like the careful mapping of surface drainage networks and the identification of moulins and other water input points, and the use of these data as inputs for the subglacial water routing model. The paper thus has the potential to be a very useful addition to the literature.

However, the paper needs a lot of work before it is ready for publication. Referee 1 has made specific recommendations for tightening up the paper structure, with which I agree. These recommendations will help to create a clearer logical progression from observations to analysis, followed by validation of results and interpretation. This will clarify the aims, results and conclusions of the paper, increasing its clarity and impact. I also agree with Referee 1 that the language requires thorough checking throughout. The grammar, spelling and sentence structure all need careful revision, ideally with the help of a native English speaker. I don't know how 'conduit' came to be spelled 'conduct' throughout the paper, but the fact that this fundamental term came to be misspelled in a paper about glacier hydrology highlights the extent of the problem.

> Author's response:

I totally agree for the language mistakes, that is why, as we have no native English involve in the paper, it will be send to a professional person to correct the language before the final submission.

> Author's changes in manuscript:

English language will be corrected.

- Comment from Referee:

For the most part, the methods are sound and the assumptions reasonable. One of the assumptions, however, is highly questionable with some implications for the reliability of the results. On p. 4, line 3 and following, it is stated: "In the accumulation area temperate ice and firn are present, allowing water percolation through the glacier body..." Using this assumption, the entire upper part of Hansbreen is defined as a WIA (p. 6, line 23; p. 9, line 28). Temperate ice was considered to be permeable by some early authors (e.g. Nye and Frank, 1973), but this is no longer the accepted view (see Fountain and Walder, 1998). The paper by Lliboutry (1971: J Glac 10, 15-29) is well worth reading on this topic. He provides detailed observational evidence and theoretical considerations that show that bulk temperate ice cannot be permeable. These considerations thus invalidate the assumption that water can directly access the bed wherever temperate ice occurs through the whole glacier thickness (such as upper Hansbreen). Instead, firn aquifers are perched above essentially impermeable ice, with the transition occurring at about 30 m depth in the European Alps. Thus inclusion of areas of temperate ice as WIAs (Fig. 3a) is thus not justified. The WIA on Hansbreen should be redrawn omitting the temperate ice zone, and the model re-run.

Author's response:

You are right.
From the previous studies (Fountain and Walder, 1998; Lliboutry, 1971 and other), water in the accumulation area percolate through the snowpack then through the firn to create a layer of saturated water at the interface warm ice and firn. This water flows at this interface and come out on the surface at the equilibrium line or reaches the englacial system thanks to crevasses in the accumulation area. Because we are not able to visualize the crevasses in the accumulation area, if they exist, and because the area situated just below the equilibrium line is considered as a water input area (large crevassed area), we will include this water in this same water input area. To summarize, we will include the water coming from the accumulation area in the water input area mapped just below the equilibrium line. In fact from the literature, it should reach the input water area either by the englacial system or by the surface which will then be directed to the glacier bed by this big water input area just below the equilibrium line.
Also thanks to a study made by Grabiec et al, 2017, we have an estimation of the water refreezing (excluding capillary water that freezes in fall) inside the firn for Hansbreen. So a big part of the water storage in the firn will be included in the new run of the model. We do not expect so many changes on the results except that subglacial channels under the accumulation area should disappear and the water volume value of the conduits in the upper part of the glacier will be a bit different. Also it will not be any scenarios with a subglacial connection with the adjacent glacier Paierlbreen. This new result will reinforce the necessity of taking into account a discrete recharge for the subglacial modelling of the heavily crevassed tidewater glaciers.

Author's changes in manuscript:

The model will be re-run for Hansbreen. Some changes will appear on the results maps and on the water input area maps for Hansbreen.

- Comment from Referee:

Independent validation of the model results is of course difficult, given that most of the drainage system below the glaciers remains unobserved. So the location of outflow points (portals and plumes) is crucial. At present, this important information is not prominent enough in the text. It should be clearly flagged up as the key test of model output, ideally in a separate subsection labelled 'Model Validation'. The location of meltwater portals and upwellings should be plotted on Figs 7 & 8, so the reader can clearly compare the predicted conduit locations with known efflux points. It is also worth noting that a similar model to the one used by the authors was successful at predicting the location of plumes in front of the tidewater glacier Kronebreen (How et al., 2017: The Cryosphere), and also indicates that the K value varies through the melt season. The fact that Scenario (5), K = 0.85 represents subglacial channels observed at Crystal and Bird Brain Caves cannot be regarded as model validation. The modelled channels appear because the caves are specified as water influx points. The existence of subglacial channels extending from moulins at these locations is certainly worth highlighting, but this ought to be early in the paper - perhaps in Section 3.2 where the observational data are introduced.
Figures 7 & 8 are very interesting, but their impact can be greatly increased by simply changing the arrangement of the panels. At present, it is very difficult to assess the result and involves much tiresome switching back and forth between caption and panels. Instead, the panels should be arranged so that the two columns show Scenarios 2 & 5, and the three rows show K = 1, 0.85 & 0. This will immediately allow readers to see how the water input and pressure assumptions influence the results.

Author's response:

You are right, the plumes locations and the outflows mapped for Hansbreen and Werenskioldbreen should be presented on the figures 7 and 8. Because those mapping were done in the past and published in other papers (Palli et al, 2003; Majchrowska et al, 2015; Grabiec et al, 2017), it will not be enough material to create a new subsection but those papers will be cited to refer to the mapping method. Also I agree with the fact that Crystal Cave and Bird Brain Cave are present because they were mapped as a water input area but there is only one scenario with discrete recharge (K=0.85) which display a subglacial channel connected to those two cave system. Therefore in our point of view, it can be used as a validation tool. We agree with the new panel configuration proposed.

Author's changes in manuscript:

Mapped outflows will be added to figures 7 and 8. Outflows locations will be discussed in more details in the discussion part. The panel of those two figure will be rearranged as proposed: the two columns show Scenarios 2 & 5, and the three rows show K = 1, 0.85 & 0.

- Comment from Referee:

On p. 8, line 9: the possibility is mentioned that additional water might be released from winter/spring storage in the snowpack. What about the opposite possibility? i.e. how much of the calculated melt might be retained in snow/firn? Does the model simply assume that all meltwater will enter supraglacial/subglacial transport, with zero surface storage? This issue is related to the erroneous attribution of the whole accumulation area of Hansbreen as a WIA. If meltwater in this zone is retained in the snowpack, instead of being immediately transferred to the bed, then this will significantly reduce the modelled water inputs to the bed possibly with major implications for the results.

In summary, this is an interesting paper with a lot of potential. I look forward to seeing a revised version that maximises the impact of the results.

> Author's response:
>
> As mentioned above, we have an estimation of the water refreezing (excluding capillary water that freezes in fall) inside the firn for the accumulation area made by Grabiec et al, 2017.
>
> Author's changes in manuscript:
>
> The storage will be taken into account in the new run of the model for Hansbreen.

**Review of \Role of discrete water recharge from supraglacial drainage systems in modeling patterns of subglacial conduits in Arctic glaciers"**

All the points of the review below were taken into account during the review. The title was changed; the Abstract and the Introduction were completely rewritten; all the other parts were reworked; figure 2 was deleted; appendix part was created; all the English language of the text was reworked. Basically, the entire article was modified.

It is not mentioned bellow but we changed **subsection 3.1** and moved a part of it in the **discussion part p17 l2-11**.

Referee 1.

**General comment**

- Comment from Referee:

This paper aims to assess the importance of supraglacial hydrology on subglacial water flow. I believe this to be a very important topic in the realm of subglacial hydrology, which could have important implications for other environments including Greenland. The paper presented here, makes use of classical formulations of hydraulic potential to define subglacial water flow paths, then by some mechanism, the model is run with discrete water inputs and a supraglacial hydrology model and the results are compared. It is possible that this simplistic approach could serve as a useful framework to quantify the effects of supraglacial hydrology on the subglacial drainage system. **However, the steps that taken to apply the different hydrology distributions to the subglacial system is not clear to me. Until this is clarified, it is not possible to judge the scientific quality of this work.**

Author's response:

All the grid cells of the subglacial model are weighted by the total amount of water penetrating the glacier or not thanks to our supraglacial drainage mapping and the modelling of the different water catchment areas.

Author's changes in manuscript:

In order to clarify the method, a new paragraph was added **p9 l6-15** in the **3.4 section** of the methodology part.

- Comment from Referee:

Many of the sentences, while mostly comprehensible, are difficult to follow. Some terminology is misleading, confusing or incorrect. The presentation of some data is very poor and not of the standard of scientific publications (significant figures…) Lastly, the manuscript is poorly organized.

It appears that Mr. Decaux is a PhD student. I believe that his advisors must do a more thorough job of preparing this manuscript and guiding Mr. Decaux; it is not the job of reviewers to do this. Should the methods prove to be viable, I believe that the results, when fully described and presented in a logical manner, presented here could be a demonstration of variability caused by the different treatments of supraglacial hydrology. The results of the paper could be interesting, however, because of the issues with organization and the lack of clarity in the methods, I do not see how this can be accepted for publication. I hope to have provided some examples and comment, which when applied to the remainder of the manuscript, can improve it in the event of another submission.

Author's response:

I totally agree for the language mistakes, that is why, as we have no native English involve in the paper, it will be send to a professional person to correct the language before the final submission.
Even if the mapping of the supraglacial system was done manually, we believe that the combination of high resolution satellite images with field observations and GPS measurements gave us a good representation of the supraglacial system of the two glaciers.

Author's changes in manuscript:

English language has been corrected in the entire article.
The **abstract** and the **introduction** have been rewritten.
A part of the last paragraph of the introduction has been moved to create the new subsection **3.5** of the **methods** part **p10**.

**Terminology**

- Comment from Referee:
  **Recharge**: This either needs a new word or to be specifically defined. Do you mean seasonal water flux? Would mean seasonal water discharge be better? Discharge sum?

  Author's response:

  **Recharge** is a term widely used in the literature, e.g.:

Benn, Douglas, et al. "Englacial drainage systems formed by hydrologically driven crevasse propagation." Journal of Glaciology 55.191 (2009): 513-523.

Siegel, D. I., and R. J. Mandle. "Isotopic evidence for glacial meltwater recharge to the Cambrian-Ordovician aquifer, north-central United States." Quaternary Research 22.3 (1984): 328-335.

Ofterdinger, U. S., et al. "Environmental isotopes as indicators for ground water recharge to fractured granite." Ground water 42.6/7 (2004): 868.

Ofterdinger, U. S., et al. "Environmental isotopes as indicators for ground water recharge to fractured granite." Ground water 42.6/7 (2004): 868.

Döll, Petra, and Kristina Fiedler. "Global-scale modeling of groundwater recharge." Hydrology and Earth System Sciences Discussions 4.6 (2007): 4069-4124.

Gulley, J. D., et al. "The effect of discrete recharge by moulins and heterogeneity in flow-path efficiency at glacier beds on subglacial hydrology." Journal of Glaciology 58.211 (2012): 926-940.

**Recharge** refers to any input of water in the system (occurring mainly during the ablation season).

Author's changes in manuscript:

**Recharge** is better defined in the **Introduction p2 l18-20**.

- Comment from Referee:

**discrete recharge/ homogeneous recharge**: I think this is misleading. Homogeneous recharge, to me, seems like putting water equally over the glacier bed. I might suggest the terms with/without supraglacial hydrology. Whatever the terminology, it must be made clear in the introduction/methodology. Sometimes the work theory is used, I think a better word would be "test case" or \scenario"

Author's response:

Yes you well understood the notion of Homogeneous recharge, it is "like putting water equally over the glacier bed" or having water put all over the glacier bed in function of the spatialized water runoff model used (like explain page 9 with the different simulation scenarios). It will be clearer to use "spatially uniform recharge" instead of "homogeneous recharge" as used by Gulley et al 2012

Author's changes in manuscript:

"Homogeneous recharge" has been change into "spatially uniform recharge" and it is defined **p2**

**l20**.

"discrete recharge" has been better defined in the **abstract p1 l4-6** and in the **introduction part p2 l26-28**.

We replaced "homogeneous water recharge theory" by "spatially uniform recharge test case".

We replaced "discrete water recharge theory" by "discrete water recharge test case".

We changed "hydraulic potential theory" into "hydraulic potential gradient" or "hydraulic potential surface".

- Comment from Referee:

  **precipitations**: This is not a word. Use precipitation.

  Author's response:

  You are right.

  Author's changes in manuscript:

  We replaced "precipitations" by "precipitation"

- Comment from Referee:

  A large point is made of these glaciers being **polythermal**. However, I found it hard to follow how this fit into the scheme presented here. Discuss the implications of it being a polythermal glacier, including in **Methods** and **Discussion** or only briefly mention in Study area.

  Author's response:

  We made a point on the fact that they are polythermal because considered cold ice as impermeable and warm ice as permeable. But as mentioned by the second referee, warm ice is not considered as permeable anymore. It will not change our results and conclusions but it will allows us to increase the impact of our article by extending our results for all glaciers (polythermal, temperate and cold) displaying a channelized supraglacial system with moulins and/or crevassed areas. So we will focus on the different surfaces properties:
  - firn = infiltration and englacial circulation.
  - bare ice (cold or warm) displaying stream on the surface = runoff.
  - crevassed area = penetration of water.

Author's changes in manuscript:

We corrected this point in the different parts of the article. We only mentioned that they are polythermal but we didn't insist on it too much on it anymore. We **deleted figure 2** and we were then able to extend our results for all Arctic glaciers which have an internal drainage system.

- Comment from Referee:

  **conducts**: I assume you mean \conduit".

  Author's response:

  You are right.

  Author's changes in manuscript:

  We replaced "conducts" by "conduits".

**Organization**

- Comment from Referee:

  **Abstract** This needs to be reworked. For improvement, I encourage you to look at resources such as this one: http://sepwww.stanford.edu/sep/prof/abscrut.h

  Author's response:

  You are right.

  Author's changes in manuscript:

  The **abstract** was rewritten.

- Comment from Referee:

  **Introduction** consists of three parts. 1) The background knowledge, broad to specific. 2) The gap in knowledge that the paper will address. 3) Your solution to the knowledge gap. Your work should only be discussed in the last paragraph or two, and should summarize what you will do. The description that takes up most of the introduction here does not belong here.

Author's response:

You are right.

Author's changes in manuscript:

The **introduction** was rewritten.
We developed more the background knowledge about the models already existing and we better precised the gap in knowledge. We also better summarized our work and focused more on how we will answer to the gap mentioned at the beginning of the introduction.
Also we moved a part of the introduction in the methodology part especially in the new subsection **3.5 p9-10**.

- Comment from Referee:

  **Page 5 Section 3.1.1.** This paragraph is way too long. Try making it just a couple of sentences describing the overall scheme. Put dates, datasets, etc. in the respective subsections. Also make this section 3.1. Consider changing the name of the previous section to **Study sites and datasets** and put the dataset description there. i.e. weather data, imagery.

  Author's response:

  You are right.

  Author's changes in manuscript:

  We moved this paragraph into the new part call **study sites and datasets p5**. We shortened this paragraph by removing all the explanations on the reason of the use of VHRS.

- Comment from Referee:

  I would encourage you to make the **descriptions of experiments** a separate subsection. Its title could be "model runs" or "experiment design".

  Author's response:

  You are right.

  Author's changes in manuscript:

  We created an extra subsection for the description of the different scenarios calls "**model runs**" **p9-10**.

- Comment from Referee:

  **Page 10 Lines 6-8** You present several different K values. However, your results only utilize a few. Either exclude the others, or make the plots/maps available in some kind of supplementary material.

Author's response:

We explain why we do not present the results for some K values page 13 lines 1-3.
The fact that the whole K values are mentioned allows us to justify why we are focusing on the three
K value K=0; K=0.85 and K=1. Therefore it seems essentials to mention them to show the logic of our
selection.

Author's changes in manuscript:
We made available 6 extra maps per glacier with discrete and homogeneous recharge for K=0.75;
K=0.5 and K=0.25 in an **appendix part p23-24**.

- Comment from Referee:

I found quite a few **Results** presented in the **Discussion** section. This is a big issue with this
paper. Separating these topics can be difficult. This needs to be reorganized, and consider if
combining the sections could be a good idea.

Author's response:

We do not agree with this statement. Here is the list of the results presented compare to the
discussion part. One thing should move from Results part to Discussion part, it is the
interpretation of different K scenarios.
Also we prefer to keep the classical way of an article organization without combining the two
sections.

Results:
- Supraglacial map description
- Interpretation of different K scenarios
- Hansbreen simulation description + no water volume but identification of main channel
- Werenskioldbreen simulation description + no water volume but identification of main
channel

Discussion:
- Supraglacial map discussion
- Discussion of Hansbreen simulation + water volume
- discussion of Werenskioldbreen simulation + water volume
- Discrete recharge/Homogeneous recharge
- Connection with Paierlbreen
- Limit of the model in water volume values

Author's changes in manuscript:

We moved from **Results** part to **Discussion** part the interpretation of different K scenarios **p17
l27-32 / p18 l1-5**.

- Comment from Referee:

  **Supraglacial drainage system evolution**. This section has some interesting little facts, however, I don't see how it is coherent to the overall story of the paper (and I am not entirely sure what the greater story is). Either find a way to make it relevant, or excluded it.

  Author's response:

  It is important because it allows us to estimate the validity duration of our model.
  This timescale evolution might be the same for all the Arctic glaciers and because we demonstrate that it plays a very important role on the location of the subglacial channels, it is crucial to analyze its evolution through the time.

  Author's changes in manuscript:

  We developed a bit more in the **discussion** part the fact that it allows us to estimate the validity duration of our model and why it is important **p17 l20-26**.

- Comment from Referee:

  **Page 13 Lines 1-12** These descriptions of K should be condensed and added to methods.

  Author's response:

  We agree that we have to move it but it should be in the discussion part because we find this classification:
  -        K=1; K=0.85 (because both of them display more than one outflow)
  -        K=0.75; K=0.5; K=0.25; K=0 (because all of them display one outflow)
  thanks to our results. Therefore, we cannot move it in to methods.

  Author's changes in manuscript:

  We moved from **Results** part to **Discussion** part the interpretation of different K scenarios **p17 l27-32 / p18 l1-5**.

- Comment from Referee:

  **Page 13 Lines 13-15** Where are these caves discussed? How are caves better represented with a discrete water recharge? This needs to be clearer

  Author's response:

  The caves are only mark on the map in study area section. You are right, it is necessary to mentioned them in the text in this section. Also, we will explain how it is better represented with K=0.85 discrete recharge scenario (one or two sentence).

Author's changes in manuscript:

We mentioned the caves in the text of **Study sites and datasets p4 l8-10**. We explain how it is better represented with K=0.85 discrete recharge scenario **p13 l10-12 / p18 l17-20**.

- Comment from Referee:

  **Page 17 Lines 10** Discuss why this is important to your work.

  Author's response:

  You are right it is developed in the conclusion page 19 lines 7-9 but not in the discussion part.

  Author's changes in manuscript:

  We developed a bit more on how the supraglacial system evolution tells about the subglacial system evolution in **Introduction part / p17 l20-26**.

- Comment from Referee:

  **Page 17 Lines 12-13, Lines 27-29** I believe most people would believe amounts of water to be a result as opposed to something to be discussed. Also what to the + and - stand for? What are the uncertainties? Standard deviations? Model errors? Variability in the input data? You present the water discharge quantity in confidence to the $m^3$. That seems like an overly precise measurement to me. This is not how numbers in are represented in science, look into significant figures .

  Author's response:

  We can oppose our total water input volume calculated for 2015 with the total runoff of all the catchment of Werenskioldbreen area measured on the field for the years 2007-2011 by Majchrowska et al., 2015. This is not possible for Hansbreen as it is a tidewater glacier. Also we can compare the data of having 68.7% of the water supply by meltwater with observations made by Majchrowska et al., 2015.  With those comparisons we know that we achieve realistic values. +/- is the calculated error of our spatialized water runoff calculation.
  You are right the values should not be present with a confidence to the $m^3$.

  Author's changes in manuscript:

  We validated our spatialized water runoff calculation by comparing Werenskioldbreen values with Majchrowska et al., 2015 values **p18 l25-33 p19 l1**.
  We explained our calculation error **p8 l1-7**.
  We changed our $m^3$ values confidence. For example water values are presented with this confidence 43.81 $10^6$ $m^3$ **p6 l23 / l25 p18 l6 / l25 / l27**.

- Comment from Referee:

  **Page 17-18** You repeatedly talk about outflow observations. What kind of data is this? Another paper (then cite)? if not, you need to make section in your **Methods/Study area** to describe how

this data was collected. This shows that your model can somehow be verified (a very good thing…). Making a point of describing this will strengthen your conclusion.

Author's response:

You are right. There were some outflows positions mapped in the past and it fit quite well with our model.

Author's changes in manuscript:

We cite Palli et al 2003 and Grabiec 2017 for Hansbreen **p18 l20-22** and Majchrowska, 2015; Palli et al 2003 and Grabiec 2017 for Werenskioldbreen **p19 l5-9** outflows publish. Finally, it fit with personal field observations from the authors.
We also added outflow positions mapped on our two results figures + two appendix figures.

- Comment from Referee:

  **Page 18 Lines 26** underestimated not under-estimating.

  Author's response:

  You are right.

  Author's changes in manuscript:

  We replaced "under-estimating" by "underestimated".

- Comment from Referee:

  **Page 20** I think you could probably end your paper on a stronger note by highlighting the difference between the two situations.

  Author's response:

  This is done page 19 lines 19-23.

  Author's changes in manuscript:

  We change the paragraph **p20 l27-31 / p21 l1-3**

**Scientific Comment**

- Comment from Referee:

  **Page 2 Lines 1-6** Why is subglacial hydrology important? Sliding? subglacial sediment transport? ice marginal lake drainages?

  Author's response:

  You are right, we mention the impact on dynamic in the abstract but not in introduction: part of the work which should be done on the introduction as mentioned before.

  Author's changes in manuscript:

  We rewrote the entire **introduction**.

- Comment from Referee:

  **Form Page 2 Line 7 to Page 3 Line 10** This needs to be rewritten, omitting what is done in your experiment, and using the formulation above.

  Author's response:

  You are right: part of the work which should be done on the introduction as mentioned before.

  Author's changes in manuscript:

  We rewrote the entire **introduction**.

- Comment from Referee:

  **Page 2 Lines 33** Many have worked on the role on non-uniform water input to hydraulic models, these include Werder et. al. 2013, Hewitt 2013, Poinar et al., 2015.

  Author's response:

  You are right: part of the work which should be done on the introduction as mentioned before.

  Author's changes in manuscript:

  We rewrote the entire **introduction**.

- Comment from Referee:

  **Methodology**, **Spatialized water runoff calculation** Shorten to "Spatially distributed runoff estimation". More importantly, it is not clear how this is done. Do you use a T-index model? Do you prescribe the discharge with an elevational gradient/hypsometry? This needs a better explanation.

Author's response:

You are right.

Author's changes in manuscript:

We renamed the section into "**Estimation of spatially distributed runoff**".
We better described the relation between summer mass balance and elevation mentioned in this section **p6 l26-31**.

- Comment from Referee:

  **Page 8 Line 17**. Where does the bed DEM come from?

  Author's response:

  The origin of the DEM of the bedrock is well described in the dataset part with in addition the reference to Grabiec et al 2012.

  Author's changes in manuscript:

  Nothing.

- Comment from Referee:

  **Methodology** You need a section detailing how the $Q_h$ is coupled to the subglacial model. The whole paper relies on this, and it is not evident to me what was done. This is one reason why I don't find the paper to be publishable.

  Author's response:

  You are right, the way how the water input data are apply to the model is explained in the description of the scenarios, page 9 lines 8-26, but for some precise case not with a general point of view.

  Author's changes in manuscript:

  We better explained the fact that we take into account all kind of precipitation **p6 l13-19**.
  We changed Qh into Qp to precise that is precipitation and not the total amount of water **p7**.
  We precised that we obtain three input files with water values:
  - Precipitation
  - Ablation
  - Precipitation + ablation

  Qm / Qp / Qh **section 3.3**

We better developed how the water input data are apply to the model in a general point of view **p9 l6-15**.

- Comment from Referee:

    **Page 10 Lines 1-2** No. Rothlisberger 1972 and Weertman 1972, discuss englacial water flow, not surface water flow.

    Author's response:

    You are right for Weertman.
    Regarding Rothlisberger, he says that the pressure in the conduits depend directly on the discharge so by the logic of the mechanisms involve, it implies that it depends on the recharge.

    Author's changes in manuscript:

    We removed Weertman 1972, we kept Rothlisberger 1972 and explain better the relation **p10 l11-14**.

- Comment from Referee:

    **Page 17 Lines 7-9** There has been plenty of work examining moulins, if you choose to discuss them, then add citations.

    Author's response:

    You are right.

    Author's changes in manuscript:

    We added citations **p17 l17-18**.

- Comment from Referee:

    **Page 18 Lines 12-14** How do you know that water does not penetrate the cold glacier ice? Do you have field observations? or did you read about this in a paper (which you then NEED to cite)?

    Author's response:

    This is well known in glaciology.

    Author's changes in manuscript:

We precised that we only speek about bare ice surface and not cold or temperate plus citation **p19 l20-23**.

- Comment from Referee:

**Page 19 Lines 20-23** How does subglacial hydrology differ between tidewater and land-terminating glaciers? How do tidewater glaciers become more crevassed? This sections need more citations and should go in the discussion section. Poinar et al., 2015 discusses the implications of surface hydrology and moulins. I would be nice to put this work in relation to that research.

Author's response:

You are right, it needs more precision.  The fact that tidewater glaciers are more crevassed is well known in glaciology. This is due to the fact that in general, tidewater glaciers have a higher dynamic than land-terminating glaciers. There is no big differences between tidewater and land-terminating glacier's subglacial hydrology system, except due to the fact that tidewater glaciers are more crevassed.

Author's changes in manuscript:

In the discussion part we developed the differences between tidewater and land-terminating glaciers and add new citations **p18 l12-15**.

- Comment from Referee:

**Page 19 Lines 24-28** Here it seems like you are trying to discuss the future of the subglacial drainage system, with the moving moulins due to the flow of ice. However, as Fischer et al. 2005 point out surface elevation changes (do to increased melt, etc.) also will affect the location of the subglacial drainage system. This is another point worth mentioning looking toward the future.

Author's response:

The fact that we based our estimation of the subglacial drainage in the future on the evolution of the supraglacial drainage system which is the result of the glacier flow and the surface elevation changes, by definition, take into account those two components. In Fisher et al 2005, the impact of the glacier geometry changes on the subglacial system are noticeable mainly due to the presence of a medial moraine which is increasing and impact the system. This is not the case for our glaciers. Even if there is a medial moraine on Werenskioldbreen, it does not impact the subglacial system as the channels are passing bellow this medial moraine (observed by Czech explorations cf their field report).

Author's changes in manuscript:

Nothing.

**Figures**

- ### Comment from Referee:

  I do not think that the coordinate system used in necessary to mention.

  ### Author's response:

  It was not mentioned before but it was a request of the editor before the submission.

  ### Author's changes in manuscript:

  We removed the coordinate system of all the figures except for the **figure 1**. We also keep it for **figure 5** because it is a different one.

- ### Comment from Referee:

  **Figure 1** Mass balance, not Masse balance. Also adding some interpreted flowlines will help the reader orient themselves.

  ### Author's response:

  You are right for mass balance. Concerning the flow direction, having the front part of the glacier visible + the topographic lines, in our point of view, it is not necessary to add some interpreted flowlines having the basic knowledge that a glacier is flowing from higher to lower elevation due to the gravity.

  ### Author's changes in manuscript:

  We changed "masse balance" for "mass balance".

- ### Comment from Referee:

  **Figures 7 and 8** What are the different columns? Mention this in the caption. Also I find the UTM markers a bit distracting and not necessary. These should be removed.

  ### Author's response:

  You are right for the UTM coordinate. There is nothing specific to the columns. It is just needed to refer to the legend and to the letter corresponding to the figure (a); b); etc…).

  ### Author's changes in manuscript:

  We removed the UTM coordinates.
  Other changes have been applied in the organization of those two figures, cf answer to the other review bellow.

This paper looks at the relationship between surface water inputs and subglacial drainage networks, an important topic that previously has received relatively little attention. I especially like the careful mapping of surface drainage networks and the identification of moulins and other water input points, and the use of these data as inputs for the subglacial water routing model. The paper thus has the potential to be a very useful addition to the literature.

However, the paper needs a lot of work before it is ready for publication. Referee 1 has made specific recommendations for tightening up the paper structure, with which I agree. These recommendations will help to create a clearer logical progression from observations to analysis, followed by validation of results and interpretation. This will clarify the aims, results and conclusions of the paper, increasing its clarity and impact. I also agree with Referee 1 that the language requires thorough checking throughout. The grammar, spelling and sentence structure all need careful revision, ideally with the help of a native English speaker. I don't know how 'conduit' came to be spelled 'conduct' throughout the paper, but the fact that this fundamental term came to be misspelled in a paper about glacier hydrology highlights the extent of the problem.

> Author's response:

I totally agree for the language mistakes, that is why, as we have no native English involve in the paper, it will be send to a professional person to correct the language before the final submission.

> Author's changes in manuscript:

English language Has been corrected for the entire article.

- Comment from Referee:

For the most part, the methods are sound and the assumptions reasonable. One of the assumptions, however, is highly questionable with some implications for the

reliability of the results. On p. 4, line 3 and following, it is stated: "In the accumulation area temperate ice and firn are present, allowing water percolation through the glacier body..." Using this assumption, the entire upper part of Hansbreen is defined as a WIA (p. 6, line 23; p. 9, line 28). Temperate ice was considered to be permeable by some early authors (e.g. Nye and Frank, 1973), but this is no longer the accepted view (see Fountain and Walder, 1998). The paper by Lliboutry (1971: J Glac 10, 15-29) is well worth reading on this topic. He provides detailed observational evidence and theoretical considerations that show that bulk temperate ice cannot be permeable. These considerations thus invalidate the assumption that water can directly access the bed wherever temperate ice occurs through the whole glacier thickness (such as upper Hansbreen). Instead, firn aquifers are perched above essentially impermeable ice, with the transition occurring at about 30 m depth in the European Alps. Thus inclusion of areas of temperate ice as WIAs (Fig. 3a) is thus not justified. The WIA on Hansbreen should be redrawn omitting the temperate ice zone, and the model re-run.

Author's response:

You are right.
From the previous studies (Fountain and Walder, 1998; Lliboutry, 1971 and other), water in the accumulation area percolate through the snowpack then through the firn to create a layer of saturated water at the interface warm ice and firn. This water flows at this interface and come out on the surface at the equilibrium line or reaches the englacial system thanks to crevasses in the accumulation area. Because we are not able to visualize the crevasses in the accumulation area, if they exist, and because the area situated just below the equilibrium line is considered as a water input area (large crevassed area), we will include this water in this same water input area. To summarize, we will include the water coming from the accumulation area in the water input area mapped just below the equilibrium line. In fact from the literature, it should reach the input water area either by the englacial system or by the surface which will then be directed to the glacier bed by this big water input area just below the equilibrium line.
Also thanks to a study made by Grabiec et al, 2017, we have an estimation of the water refreezing (excluding capillary water that freezes in fall) inside the firn for Hansbreen. So a big part of the water storage in the firn will be included in the new run of the model. We do not expect so many changes on the results except that subglacial channels under the accumulation area should disappear and the water volume value of the conduits in the upper part of the glacier will be a bit different. Also it will not be any scenarios with a subglacial connection with the adjacent glacier Paierlbreen. This new result will reinforce the necessity of taking into account a discrete recharge for the subglacial modelling of the heavily crevassed tidewater glaciers.

Author's changes in manuscript:

The model has been re-run for Hansbreen as explained in all the different part of the article, we don't allow penetration of water in temperate ice anymore but just an infiltration which goes down to some WIAs.
There are some changes on the results maps and on the water input area maps for Hansbreen **Figure 2 (a) / 4 (a) / 6 (d) (e) (f)**.

- Comment from Referee:

Independent validation of the model results is of course difficult, given that most of the drainage system below the glaciers remains unobserved. So the location of outflow points (portals and plumes) is crucial. At present, this important information is not prominent enough in the text. It should be clearly flagged up as the key test of model output, ideally in a separate subsection labelled 'Model Validation'. The location of meltwater portals and upwellings should be plotted on Figs 7 & 8, so the reader can clearly compare the predicted conduit locations with known efflux points. It is also worth noting that a similar model to the one used by the authors was successful at predicting the location of plumes in front of the tidewater glacier Kronebreen (How et al., 2017: The Cryosphere), and also indicates that the K value varies through the melt season. The fact that Scenario (5), K = 0.85 represents subglacial channels observed at Crystal and Bird Brain Caves cannot be regarded as model validation. The modelled channels appear because the caves are specified as water influx points. The existence of subglacial channels extending from moulins at these locations is certainly worth highlighting, but this ought to be early in the paper - perhaps in Section 3.2 where the observational data are introduced.
Figures 7 & 8 are very interesting, but their impact can be greatly increased by simply changing the arrangement of the panels. At present, it is very difficult to assess the result and involves much tiresome switching back and forth between caption and panels. Instead, the panels should be arranged so that the two columns show Scenarios 2 & 5, and the three rows show K = 1, 0.85 & 0. This will immediately allow readers to see how the water input and pressure assumptions influence the results.

  Author's response:

  You are right, the plumes locations and the outflows mapped for Hansbreen and Werenskioldbreen should be presented on the figures 7 and 8. Because those mapping were done in the past and published in other papers (Palli et al, 2003; Majchrowska et al, 2015; Grabiec et al, 2017), it will not be enough material to create a new subsection but those papers will be cited to refer to the mapping method. Also I agree with the fact that Crystal Cave and Bird Brain Cave are present because they were mapped as a water input area but there is only one scenario with discrete recharge (K=0.85) which display a subglacial channel connected to those two cave system. Therefore in our point of view, it can be used as a validation tool. We agree with the new panel configuration proposed.

  Author's changes in manuscript:

  Mapped outflows have been added to **figures 7 and 8 (+ appendix)**.
  Outflows locations have been discussed in more details in the **discussion part p18 l17-23 + p19 l5-9**.
  The panel of those **figure 6 / 7 (+appendix)** have been rearranged as proposed: two columns show Scenarios 2 & 5, and the three rows show different K values.

- Comment from Referee:

On p. 8, line 9: the possibility is mentioned that additional water might be released from winter/spring storage in the snowpack.  What about the opposite possibility?  i.e.  how much of the calculated melt might be retained in snow/firn?  Does the model simply assume that all meltwater will enter supraglacial/subglacial transport, with zero surface storage?  This issue is related to the erroneous attribution of the whole accumulation area of Hansbreen as a WIA. If meltwater in this zone is retained in the snowpack, instead of being immediately transferred to the bed, then this will significantly reduce the modelled water inputs to the bed possibly with major implications for the results.

In summary, this is an interesting paper with a lot of potential. I look forward to seeing a revised version that maximises the impact of the results.

> Author's response:

> As mentioned above, we have an estimation of the water refreezing (excluding capillary water that freezes in fall) inside the firn for the accumulation area made by Grabiec et al, 2017.

> Author's changes in manuscript:

> The storage has been taken into account in the new run of the model for Hansbreen as explained **p6 l19-25**.

---

## Referee Report (RR1)

**Review of**
*Role of discrete water recharge from supraglacial drainage systems in modeling patterns of subglacial conduits in Arctic glaciers*
by Decaux et al.

**General comments**
This manuscript presents calculated locations and water fluxes through subglacial channels under two Svalbard glaciers. It compares a "spatially uniform recharge" scenario, in which melt and rain water is allowed to enter the subglacial system locally, to a "discrete recharge" scenario, in which the water may only enter at identified moulin or crevasse locations. The study finds better agreement between modeled and observed locations of subglacial outflow when the "discrete recharge" scenario is used.

The result is important and reflects conclusions of other recent work that couples surface hydrologic networks to subglacial hydrology models (e.g., Banwell et al. (2013), Gulley et al. (2012), Bartholomew et al. (2011), to name just a few).

The manuscript too frequently overstates claims, makes assumptions without evidence, lacks presentation of field (or remote sensing) observations that support or refute their predicted subglacial conduit locations and fluxes, and suffers from an imprecise writing style. If these shortcomings can be addressed, it could merit publication in The Cryosphere.

**Specific comments**
P1 L6-8 Most current subglacial hydrology models DO include heterogeneous recharge. Unweighted hydropotential flow accumulation calculations are still regularly performed, but I would no longer consider this the "standard model". I suggest rephrasing this sentence accordingly.

P1 L15, 18 The results are generalized to "Arctic tidewater glacierS" and "land-terminating glacierS", yet only one of each type was studied, without placing them into any context of being typical or atypical of other Arctic glaciers. This generalization needs to be either supported or removed.

P1 L20 The predicted conduits are not compared to observations; therefore, "more realistic results" here is not supported.

P1 L20-21 The meaning of this sentence is unclear and should be removed or reworded.

P2 L33-34 "no [model] has used a real representation of the supraglacial drainage system" is patently false. Banwell et al. (2013), Colgan et al. (2011), Mayaud et al. (2014), Bougamont et al. (2014) are studies that have done this.

P3 L1-20 This three-paragraph summary of the manuscript does not belong in the Introduction. If you must outline your paper here, limit yourself to 3-4 sentences at the most.

P3 L23 This statement needs citation.

P3 L32 Ryser et al. (2013) is a natural citation for this statement.

P5 L14-19 This section on the unsuccessful application of automated stream detection algorithm should either be enhanced – stating more detail about the broadband overlap in reflectance, possibly including a comparison of debris-covered and relatively debris-free regions – or removed.

P7 L1 What is the origin of $\Delta_P = 19\%$? This should be cited and briefly explained.

P7 L4 All the subscript in this equation make it difficult to read. You could consider using $Q_0$ for the amount of precipitation at your AWS station, since it is sited at roughly 0 meters a.s.l.

P8 L4 Errors in $\Delta_P = 19\%$ are not accounted for here. I suspect these will be larger than errors in $h$ or $Q_{PPS}$ due to expected substantial meteorological variations between rainfall events. It may be difficult to know and quantify such errors, so at the very least this additional uncertainty should be commented on.

P8 L7 Usually errors are added in quadrature.

P9 L17 - P10 L8 Five scenarios are described, but results from only two scenarios (#2 and #5) are presented. I suggest removing the other three scenarios, which will simplify the presentation.

P9 L22-24 I would not refer to Scenario #2 as "spatially uniform recharge" since water input is allowed to vary spatially according to local production at the surface (Figure 3). Instead, you might call it a "local recharge" scenario, or something like that, to describe the lack of surface meltwater routing.

P10 L16-18 Artesian features (which is a more precise way to say "geyser-like spouts of water") suggest $k > 1$. Although rather nonstandard, you might consider adding $k > 1$ for Werenskioldbreen; Everett et al. (2016) have done this for a Greenland glacier.

P10 L26-27 Are the locations of the main subglacial channels somehow seeded by the authors in their model? Presumably they originate at locations of concentrated recharge, but this sentence suggests they might be baked into the model. Clarify.

P11 A new subsection to include methods of field or remote observations of subglacial conduits needs to be added.

P12 L2 Subsection heading: What does "Changes" refer to – changes over time, space, due to model scenario, etc.? Clarify.

P13 A new subsection discussing the goodness of fit of field observations to the predicted subglacial conduit locations and fluxes needs to be added. Relevant parts of the "authors' personal unpublished maps" must be included here.

P14 L13 How does the current approach and the results differ from those of P alli et al (2003)?

P17 L21 The assumption that observations from these two glaciers in 1990, 2010, and 2011 can be "extrapolated to the entire Arctic" is terribly overblown.

P17 L24-26 Here it is noted that "few changes" were found between 1990-2010, yet in the Results

section (4.1), "several changes" were noted, grouped into four broad classes. This inconsistency must be addressed before you can claim that your results will be valid on decadal timescales.

P18 L6, L25 These water volumes are very precise. At least one significant figure should be dropped, if not two.

P18 L18-20 The subglacial channels mentioned here are not generally "well known"; any data used to identify such channels needs to be included in the manuscript.

P19 L14-18 Why is this important?

P20 L7-8 This assumption is not adequately supported.

P20 L15-16 This was not tested or shown in the study. The subglacial flow accumulation algorithm was run on glacier geometry (surface DEM) dating to 2015. Flow accumulation at other time periods was not assessed.

P20 L28 There is no reason that I know of that subglacial channels cannot form underneath an accumulation zone.

P21 L4-8 This statement directly contradicts that on P20 L15-16 (which, as I noted above, has its own issues). Regardless of which may be true, they are not constrained by this study. If Grabiec et al. (2017) have results that would support one of these statements, they should be described here and then folded into these points.

P20-21 The conclusion section is far too long. It does not need two paragraphs to restate results, and it certainly should not refer to specific figures. I would start by deleting the first three paragraphs, then winnowing the final three paragraphs into 10-15 lines.

Figure 6: The two caves should be noted on these maps as well (red dots would be sufficient).

Appendix: I do not think these figures are necessary.

**Technical corrections**
P2 L20 Mistakenly written "heterogeneous" instead of "homogeneous"

P2 L28-30 This is true for temperate glaciers

P2 L31 Smith et al. (2015) would be ideal to cite in support of this sentence

P6 L15 "spatialized" is not a word

P6 L27 WGMS should be written out and a citation added

P10 L22 Specify "supraglacial" drainage catchment structure

P10 L25 Specify that this refers to $k$

P12 L22 Specify "more" consistent "than the 1990-2010 pair"; they are not fully consistent, just

more consistent than the 1990-2010 comparison

P19 L21 Absent a crevasse, moulin, conduit, or hydrofracture, this statement can be true; as written, it is not true

P19 L29, P20 L1 Use of the word "satisfying": it is not appropriate to describe emotions associated with obtaining certain results

P19 L34 If $k \leq 1$, then the subglacial system is never "overpressurized"

P20 L11 I find a factor of 3 here, not an order of magnitude.

P21 L9-11 Specify "on these glaciers" at the end of this sentence. The method is not new, but its application to Hansbreen and Werenskioldbreen is.

---

## Referee Report (RR2)

[referee-annotated manuscript omitted]

---

## Author Response (AR2)

**Review of**
*Role of discrete water recharge from supraglacial drainage systems in modeling patterns of subglacial conduits in Arctic glaciers*
by Decaux et al.

**General comments**

- Comment from Referee:

This manuscript presents calculated locations and water fluxes through subglacial channels under two Svalbard glaciers. It compares a "spatially uniform recharge" scenario, in which melt and rain water is allowed to enter the subglacial system locally, to a "discrete recharge" scenario, in which the water may only enter at identified moulin or crevasse locations. The study finds better agreement between modeled and observed locations of subglacial outflow when the "discrete recharge" scenario is used.

The result is important and reflects conclusions of other recent work that couples surface hydrologic networks to subglacial hydrology models (e.g., Banwell et al. (2013), Gulley et al. (2012), Bartholomew et al. (2011), to name just a few).

The manuscript too frequently overstates claims, makes assumptions without evidence, lacks presentation of field (or remote sensing) observations that support or refute their predicted subglacial conduit locations and fluxes, and suffers from an imprecise writing style. If these shortcomings can be addressed, it could merit publication in The Cryosphere.

Author's response:

Thanks for endorsing the study like previous reviewer.
We generally agree with your comments and will try to make the article more precise.
Nevertheless, due to harsh condition (polar night, meteorological conditions) and dealing with tidewater glacier it is impossible to have very detail evidences of subglacial channel network.

Author's changes in manuscript:

We answered and changed the article in function of this review.

**Speciftc comments**

- Comment from Referee:

P1 L6-8 Most current subglacial hydrology models DO include heterogeneous recharge. Unweighted hydropotential flow accumulation calculations are still regularly performed, but I would no longer consider this the "standard model". I suggest rephrasing this sentence accordingly.

Author's response:

We agree, several studies on the past years state on the importance to consider the supraglacial drainage system. Nevertheless, none of them made a model with complete "real" supraglacial drainage system with locations of glacier moulins and crevasses area as collectors of water and also none made the comparison with and without considering it.

Author's changes in manuscript:

We reformulated accordingly and we removed all "standard model" from the article.

-     Comment from Referee:

P1 L15, 18 The results are generalized to "Arctic tidewater glacierS" and "land-terminating glacierS", yet only one of each type was studied, without placing them into any context of being typical or atypical of other Arctic glaciers. This generalization needs to be either supported or removed.

Author's response:

We agree we need to more support the fact that they are representative of "Svalbard glaciers" and not "Arctic glaciers".

Author's changes in manuscript:

We changed "Arctic glaciers" for "Svalbard glaciers" and we developed more, in the "Study sites" section, the fact that they are both representative of Svalbard glaciers regarding their morphology and hydrothermal structure.

-     Comment from Referee:

P1 L20 The predicted conduits are not compared to observations; therefore, "more realistic results" here is not supported.

Author's response:

The predicted conduits are not compared to direct observations because it is impossible to penetrate and to map the entire englacial and subglacial conduits. Nevertheless it is possible to assess our results thanks to the observed outflow positions and some known subglacial / englacial channels location thanks to Bird Brain and Crystal caves. Also, the fact that they do not display subglacial channels in the accumulation area fit with previous theoretical studies (Fountain and Walder, 1998; Lliboutry, 1971 and other). I fact, they showed that either water in the accumulation area percolate through the snowpack then through the firn to create a layer of saturated water at the interface warm ice / firn to appear supraglacially at the equilibrium line, either it flows englacially, thanks to the presence of crevasses under the snow pack in the accumulation area, to reach the ablation area before to be redirected subglacially. Therefore, there might have some distributed inefficient drainage system below the accumulation area but no a well channelized efficient system.
Finally it could have been possible to make some drillings in order to try to asses our model, but even if we would have the field power it would not be easy due to the national park status of the area. Moreover, hot water drillings have to be very closely spaced to hit a channel and they can easily modify or even spoil natural drainage system.

Author's changes in manuscript:

We better specified in the text.

-     Comment from Referee:

P1 L20-21 The meaning of this sentence is unclear and should be removed or reworded.

Author's response:

We agree.

Author's changes in manuscript:

We removed this sentence and adapt it P1 L14-17.

•      Comment from Referee:

P2 L33-34 "no [model] has used a real representation of the supraglacial drainage system" is patently false. Banwell et al. (2013), Colgan et al. (2011), Mayaud et al. (2014), Bougamont et al. (2014) are studies that have done  this.

Author's response:

You are right some works have been done regarding Greenland, on the influence of drainage surface lakes and crevasses areas on local velocity (Bougamont et al. 2014; Colgan et al. 2011). Also some models with discrete moulin input were realized but with moulin's location not based on mapping technics but on localization of depressions areas (Banwell et al. 2013; Mayaud et al. 2014).
All those works show the importance of considering the supraglacial drainage system into the glacial hydrological models. But no assessment of the importance of using discrete or spatially uniform water recharge was realized.

Author's changes in manuscript:

We reformulated this sentence according to previous comments.

•      Comment from Referee:

P3 L1-20 This three-paragraph summary of the manuscript does not belong in the Introduction. If you must outline your paper here, limit yourself to 3-4 sentences at the most.

Author's response:

We had this previous comment from previous referee:
**"Introduction** consists of three parts. 1) The background knowledge, broad to specific. 2) The gap in knowledge that the paper will address. 3) Your solution to the knowledge gap. Your work should only be discussed in the last paragraph or two, and should summarize what you will do. The description that takes up most of the introduction here does not belong here."

We agree that it is too long but we will not shorten it to 3-4 sentences which is too short to introduce our work. Thus we will keep the format requested by the first reviewer : "work should only be discussed in the last paragraph or two, and should summarize what you will do."

Author's changes in manuscript:

We shorten this last part of the introduction.

•      Comment from Referee:

P3 L23 This statement needs  citation.

Author's response:

We agree.

Author's changes in manuscript:

We developed a bit more the statement an added citations:
(Grabiec et al., 2012; Hagen et al., 1993, 2003; Ignatiuk et al., 2014)

•      Comment from Referee:

P3 L32 Ryser et al. (2013) is a natural citation for this statement.

Author's response:

We agree.

Author's changes in manuscript:

We had Ryser et al. (2013) as a citation.

- Comment from Referee:

P5 L14-19 This section on the unsuccessful application of automated stream detection algorithm should either be enhanced – stating more detail about the broadband overlap in reflectance, possibly including a comparison of debris-covered and relatively debris-free regions – or removed.

Author's response:

There are more details about the broadband overlap in reflectance in the discussion part P17 L2-11.

Author's changes in manuscript:

We removed this section to only develop it in the discussion part.

- Comment from Referee:

P7 L1 What is the origin of $\Delta_P = 19\%$? This should be cited and briefly explained.

Author's response:

It is cited: Nowak and Hodson (2013) but we agree the sentence is not clear.

Author's changes in manuscript:

We reworked the sentence.

- Comment from Referee:

P7 L4 All the subscript in this equation make it difficult to read. You could consider using $Q_0$ for the amount of precipitation at your AWS station, since it is sited at roughly 0 meters a.s.l.

Author's response:

We agree.

Author's changes in manuscript:

We change $Q_{\mathrm{pps}}$ for $Q_0$ in the whole article.

- Comment from Referee:

P8 L4 Errors in $\Delta_P = 19\%$ are not accounted for here. I suspect these will be larger than errors in $h$ or $Q_{P\,P\,S}$ due to expected substantial meteorological variations between rainfall events. It may be difficult to know and quantify such errors, so at the very least this additional uncertainty should be commented on.

Author's response:

Nowak and Hodson (2013) estimated mean error of calculated runoff as 4% (for $\Delta p$ = 19%). Unfortunately, it is impossible to quantify $\Delta p$ errors without additional data from Nowak and Hodson (2013) modelling.
We are aware that our errors of the spatial distribution of the precipitation model could be underestimated.
However, our total glacier runoff error is also around 3%.

Author's changes in manuscript:

We added two sentences after P8L7:
"We are aware that the error of precipitation spatial distribution is possibly larger due to expected substantial meteorological variations between rainfall events. However, calculated total glacier runoff error correspond with Nowak and Hodson (2013) estimations."

•        Comment from Referee:

 P8 L7 Usually errors are added in quadrature.

Author's response:

We agree.

Author's changes in manuscript:

We changed the formula and recalculated the errors.

•        Comment from Referee:

 P9 L17 - P10 L8 Five scenarios are described, but results from only two scenarios (#2 and #5) are presented. I suggest removing the other three scenarios, which will simplify the presentation.

Author's response:

We agree.

Author's changes in manuscript:

We changed scenario 2 in 1 and scenario 5 in 2.

We removed the three scenarios 1; 3; 4 and change the rest of the text according to it.

Change scenario 2 and 5 of the figures!!!

•        Comment from Referee:

 P9 L22-24 I would not refer to Scenario #2 as "spatially uniform recharge" since water input is allowed to vary spatially according to local production at the surface (Figure 3). Instead, you might call it a "local recharge" scenario, or something like that, to describe the lack of surface meltwater routing.

Author's response:

We agree "spatially uniform recharge" refers better to scenario 1.
"local recharge" was a good idea but after using it seems to be confusing with "discrete recharge" thus we decided to use "spatial recharge".

Author's changes in manuscript:

We change the name in all the text.

-     Comment from Referee:

P10 L16-18 Artesian features (which is a more precise way to say "geyser-like spouts of water") suggest $k > 1$. Although rather nonstandard, you might consider adding $k > 1$ for Werenskioldbreen; Everett et al. (2016) have done this for a Greenland glacier.

Author's response:

We agree we change "geyser-like spouts of water" for "Artesian features".
Regarding the modeling of a scenario k>1, as it is a "non-usual case" (Everett et al. 2016) and that it happen only locally and only on a short period of time (Baranowski 1977) we decided to not implement it.

Author's changes in manuscript:

We changed "geyser-like spouts of water" for "Artesian features".

-     Comment from Referee:

P10 L26-27 Are the locations of the main subglacial channels somehow seeded by the authors in their model? Presumably they originate at locations of concentrated recharge, but this sentence suggests they might be baked into the model. Clarify.

Author's response:

No, no locations of main subglacial channels are seeded. It was just to express that the model involves the channelized system and not distributed system as mentioned below.

Author's changes in manuscript:

We removed this (v) point.

-     Comment from Referee:

P11 A new subsection to include methods of field or remote observations of subglacial conduits needs to be added.

Author's response:

We don't see the point of this new section as we don't use any direct observation of subglacial conduits in the article. While it is question of those ice cave, citations are added and all mapping methods are described in those article.

Author's changes in manuscript:

Nothing.

-     Comment from Referee:

P12 L2 Subsection heading: What does "Changes" refer to – changes over time, space, due to model scenario, etc.? Clarify.

Author's response:

We agree it needs to be clarify

Author's changes in manuscript:

We changed it for "Temporal changes"

-      Comment from Referee:

P13 A new subsection discussing the goodness of fit of field observations to the predicted sub-glacial conduit locations and fluxes needs to be added. Relevant parts of the "authors' personal unpublished maps" must be included here.

Author's response:

The personal data collected do not bring more information than the existing data already published and cited in the article. The authors have the opportunity to visit those cave systems several times a year since few years. Those repeated observations just confirm that the data cited are still valid.

Author's changes in manuscript:

P18 L19 we changed "authors' personal unpublished maps" into "authors' personal observations"

-      Comment from Referee:

P14 L13 How does the current approach and the results differ from those of Palli et al (2003)?

Author's response:

P2 L16-21 / P3 L16-18 / P8 L18-21 / P9 L18-21: We already explained that Pallis'model is based on hydrological potential gradient, that it does not take into account the supraglacial system and used a spatially uniform recharge of water.

Author's changes in manuscript:

Nothing.

-      Comment from Referee:

P17 L21 The assumption that observations from these two glaciers in 1990, 2010, and 2011 can be "extrapolated to the entire Arctic" is terribly overblown.

Author's response:

We agree.

Author's changes in manuscript:

We deleted the sentence P17 L21-22

-      Comment from Referee:

P17 L24-26 Here it is noted that "few changes" were found between 1990-2010, yet in the Results section (4.1), "several changes" were noted, grouped into four broad classes. This inconsistency must be addressed before you can claim that your results will be valid on decadal timescales.

Author's response:

We agree it is not consistent.
Our finding shows that there is some supraglacial evolution on decadal timescale but those changes does not represent a complete reorganization of the system (or WIA). The new WIA either stay more or less in the same area (about 300 m² so on a glacier and our model scale it does not change so much) or stay on the same subglacial channel axes. Especially abandoned moulins which see the creation of new upstream moulins.

Author's changes in manuscript:

We explain the situation as above by developing paragraphs in discussion and conclusion sections.

• Comment from Referee:

P18 L6, L25 These water volumes are very precise. At least one significant figure should be dropped, if not two.

Author's response:

We agree.

Author's changes in manuscript:

We removed one significant number.

• Comment from Referee:

P18 L18-20 The subglacial channels mentioned here are not generally "well known"; any data used to identify such channels needs to be included in the manuscript.

Author's response:

We agree with "well-known" formulation remark.
Regarding the fact to include more data: we refer to Mankoff 2017 and Benn 2009 maps and say that our subglacial channels modelled match with their orientations. We do not see the need to add those already published maps into the article. In fact, it will overload the article with figures especially that this topic is not the main point of this article.

Author's changes in manuscript:

We removed "well-known" and change it for "well-studied", we also added more citations to show that those systems are studied since a long time.

• Comment from Referee:

P19 L14-18 Why is this important?

Author's response:

It adds some information on the pattern of the recharge of water on the glacier and show that it is heterogeneous on the glacier surface.

Author's changes in manuscript:

We added one sentence to specify the interest.

- **Comment from Referee:**

P20 L7-8 This assumption is not adequately supported.

Author's response:

We agree it needs citations.

Author's changes in manuscript:

We added citations [Cogley et al. (2011); Hock (2005); Irvine-Fynn et al. (2011); Jansson et al. (2003)] and developed a bit more the text.

- **Comment from Referee:**

P20 L15-16 This was not tested or shown in the study. The subglacial flow accumulation algorithm was run on glacier geometry (surface DEM) dating to 2015. Flow accumulation at other time periods was not assessed.

Author's response:

We agree it is not directly tested in the article. Nevertheless during the entire article we insist on the dependency of the subglacial drainage system on the supraglacial one. Also, knowing we have some changes on a decadal timescale implies some changes in the subglacial system.

Author's changes in manuscript:

We reformulated.

- **Comment from Referee:**

P20 L28 There is no reason that I know of that subglacial channels cannot form underneath an accumulation zone.

Author's response:

Like explain in previous comment and here we cite two articles:
The fact that they do not display subglacial channels in the accumulation area is a more realistic result. In fact, previous studies (Fountain and Walder, 1998; Lliboutry, 1971 and other) show that either water in the accumulation area percolate through the snowpack then through the firn to create a layer of saturated water at the interface warm ice and firn to appear supraglacially at the equilibrium line, either it flows englacially, thanks to the presence of crevasses under the snow pack in the accumulation area, to reach the ablation area before to be redirected subglacially. So from previous theoretical studies, we can only find englacial channels in the accumulation area. Maybe it is possible to have some subglacial channels in the accumulation area if water enters directly subglacially at an interface mountain slope / glacier edge. But we have no observation of such a phenomenon here. Also some subglacial channels in accumulation area can exist due to geothermal or frictional melt but not formed by surface water and they would rather be small and inefficient.

Author's changes in manuscript:

We reformulated the sentence.

- Comment from Referee:

P21 L4-8 This statement directly contradicts that on P20 L15-16 (which, as I noted above, has its own issues). Regardless of which may be true, they are not constrained by this study. If Grabiec et al. (2017) have results that would support one of these statements, they should be described here and then folded into these points.

Author's response:

We agree.
As explain above, the model can be consider as valid for a minimum period of 20 years even if some changes of the supraglacial system were observed on a decadal timescale.

Author's changes in manuscript:

We reformulate both sections in order to be consistent.
Grabiec et al. 2017 was moved in the discussion part.

- Comment from Referee:

P20-21 The conclusion section is far too long. It does not need two paragraphs to restate results, and it certainly should not refer to specific figures. I would start by deleting the first three paragraphs, then winnowing the final three paragraphs into 10-15 lines.

Author's response:

We agree that it does not need two paragraphs to restate our results and that it should not refer to specific figures. We also agree that it is too long, nevertheless only 10-15 lines seems very short for our conclusions.

Author's changes in manuscript:

We removed all references to figures and keep only one reference to article (which is needed).
Also we shorten the whole conclusion.

- Comment from Referee:

Figure 6: The two caves should be noted on these maps as well (red dots would be sufficient).

Author's response:

We agree

Author's changes in manuscript:

We added the caves on Figure 6 with red dots.

- Comment from Referee:

Appendix: I do not think these figures are necessary.

Author's response:

Those figures were asked by the previous referees and were not present before. It allows to show all the scenarios modelled and to justify our decision to not discuss them in the paper.

Author's changes in manuscript:

Nothing.

**Technical corrections**

- Comment from Referee:

P2 L20 Mistakenly written "heterogeneous" instead of "homogeneous"

Author's response:

We agree.

Author's changes in manuscript:

We wrote "homogeneous" instead of "heterogeneous".

- Comment from Referee:

P2 L28-30 This is true for temperate glaciers

Author's response:

No, it is also true for polythermal glaciers cf citation P2 L28-30:

- Gulley, J., Benn, D., Müller, D., and Luckman, A.: A cut-and-closure origin for englacial conduits in uncrevassed regions of polythermal glaciers, Journal of Glaciology, 55, 66–80, 2009.
- Irvine-Fynn, T. D., Hodson, A. J., Moorman, B. J., Vatne, G., and Hubbard, A. L.: Polythermal glacier hydrology: A review, Reviews of Geophysics, 49, 2011.

Author's changes in manuscript:

Nothing.

- Comment from Referee:

P2 L31 Smith et al. (2015) would be ideal to cite in support of this sentence

Author's response:

We agree.

Author's changes in manuscript:

We added Smith et al. (2015).

•        Comment from Referee:

P6 L15 "spatialized" is not a word

Author's response:

We agree and notice that we used this word several times.

Author's changes in manuscript:

We change the formulation of "spatialized" in the whole article.

•        Comment from Referee:

P6 L27 WGMS should be written out and a citation added

Author's response:

We agree.

Author's changes in manuscript:

We added the citation and WGMS has been written out.

•        Comment from Referee:

P10 L22 Specify "supraglacial" drainage catchment structure

Author's response:

We agree.

Author's changes in manuscript:

We added "supraglacial".

-     Comment from Referee:

P10 L25 Specify that this refers to *k*.

Author's response:

We agree.

Author's changes in manuscript:

We added "K".

-     Comment from Referee:

P12 L22 Specify "more" consistent "than the 1990-2010 pair"; they are not fully consistent, just more consistent than the 1990-2010 comparison.

Author's response:

We agree.

Author's changes in manuscript:

We added "more".

-     Comment from Referee:

P19 L21 Absent a crevasse, moulin, conduit, or hydrofracture, this statement can be true; as written, it is not true

Author's response:

We agree but we will only speak about crevasses and moulins because they are the key elements to the water to penetrate the glacier surface. The conduits are the extension of moulins and crevasses and hydrofracture is one of the formation processes of those features.

Author's changes in manuscript:

We reformulated.

-     Comment from Referee:

P19 L29, P20 L1 Use of the word "satisfying": it is not appropriate to describe emotions associated with obtaining certain results.

Author's response:

We agree.

Author's changes in manuscript:

We reformulated.

-     Comment from Referee:

P19   L34   If   $k$   $\leq$   1,   then   the   subglacial   system   is   never "overpressurized".

Author's response:

We agree.

Author's changes in manuscript:

We changed for "higher water pressure".

-     Comment from Referee:

P20 L11 I find a factor of 3 here, not an order of magnitude.

Author's response:

We agree.

Author's changes in manuscript:

We changed for "a factor of three".

-     Comment from Referee:

P21 L9-11 Specify "on these glaciers" at the end of this sentence. The method is not new, but its application to Hansbreen and Werenskioldbreen is.

Author's response:

We agree that the idea of using a discrete recharge is not new. Several studies suggest that it might be important to consider it but none of them asses it. Moreover no study (to our knowledge) modeled the subglacial channels of a glacier using a discrete recharge based on the mapping of the moulins and crevasses areas, for the entire glacier surface, combined with modeled water volumes of each WIAs thanks to the determination of their respective water catchments.

Author's changes in manuscript:

We added "for the entire glacier surface" at the sentence to express our previous explanations.

[revised manuscript text omitted]